



# Cloud feedbacks in extratropical cyclones: insight from long-term satellite data and high-resolution global simulations

Daniel T. McCoy[1], Paul R. Field[1,2], Gregory S. Elsaesser[3], Alejandro Bodas-Salcedo[2], Brian H. Kahn[4], Mark D. Zelinka[5], Chihiro Kodama[6], Thorsten Mauritsen[7], Benoit Vanniere[8], Malcolm Roberts[2], Pier L. Vidale[8], David Saint-Martin[9], Aurore Voldoire[9], Rein Haarsma[10], Adrian Hill[2], Ben Shipway[2], Jonathan Wilkinson[2]

[1]Institute of Climate and Atmospheric Sciences, University of Leeds, UK
[2]Met Office, UK
[3]Department of Applied Physics and Applied Mathematics, Columbia University and NASA Goddard Institute for Space Studies, New York, NY, USA
[4]Jet Propulsion Laboratory, California Institute of Technology, Pasadena, CA, USA
[5]Cloud Processes Research and Modeling Group, Lawrence Livermore National Laboratory, Livermore, California, USA
[6]Japan Agency for Marine-Earth Science and Technology, Yokohama, Japan
[7]Max Planck Institute for Meteorology, Hamburg, Germany
[8]National Centre for Atmospheric Science-Climate, Department of Meteorology, University of Reading, Reading, UK
[9]Centre National de Recherches Météorologiques (CNRM), Météo-France/CNRS, 42 Avenue Gaspard Coriolis, 31057 Toulouse, France
[10]Royal Netherlands Meteorological Institute, De Bilt, Netherlands

*Correspondence to*: Daniel T. McCoy (d.t.mccoy@leeds.ac.uk)

**Abstract.** Extratropical cyclones provide a unique set of challenges and opportunities in understanding variability in cloudiness over the extratropics (poleward of 30°). We can gain insight into the shortwave cloud feedback from examining cyclone variability. Here we contrast global climate models (GCMs) with horizontal resolutions from 7 km up to hundreds of kilometers with Multi-Sensor Advanced Climatology Liquid Water Path (MAC-LWP) microwave observations of cyclone properties from the period 1992-2015. We find that inter-cyclone variability in both observations and models is strongly driven by moisture flux along the cyclone's warm conveyor belt (WCB). Stronger WCB moisture flux enhances liquid water path (LWP) within cyclones. This relationship is replicated in GCMs, although its strength varies substantially across models. In the southern hemisphere (SH) oceans 28-42% of the observed interannual variability in cyclone LWP may be explained by WCB moisture flux variability. This relationship is used to propose two cloud feedbacks acting within extratropical cyclones: a negative feedback driven by Clausius-Clapeyron increasing water vapor path (WVP), which enhances the amount of water vapor available to be fluxed into the cyclone; and a feedback moderated by changes in the life cycle and vorticity of cyclones under warming, which changes the rate at which existing moisture is imported into the cyclone. We show that changes in moisture flux drive can explain the observed trend in Southern Ocean cyclone LWP over the last two decades. Transient warming simulations show that the majority of the change in cyclone LWP can be explained by changes in WCB moisture flux, as opposed to changes in cloud phase. The variability within cyclone composites is examined to understand what cyclonic regimes the mixed phase cloud feedback is relevant to. At a fixed WCB moisture flux cyclone LWP increases with increasing



SST in the half of the composite poleward of the low and decreases in the half equatorward of the low in both GCMs and observations. Cloud-top phase partitioning observed by the Atmospheric Infrared Sounder (AIRS) indicates that phase transitions may be driving increases in LWP in the poleward half of cyclones.

## 1 Introduction

Constraining the brightening or dimming of clouds in response to warming is key to offering a more accurate prediction of 21st century climate change. Caldwell et al. (2016) showed that uncertainty in shortwave cloud feedback represented the largest contribution to uncertainty in climate sensitivity in the fifth climate model intercomparison project (CMIP5) generation of models. Model uncertainty in the shortwave cloud feedback is driven by differences in the representation of clouds in the planetary boundary layer, which contribute strongly to albedo, but not to outgoing longwave radiation (Hartmann and Short,
1980). These clouds exist at a time- and length-scale that is much finer than even the highest resolution simulation and are thus parameterized, leading to substantial disagreement in feedback from one model to another.

The shortwave cloud feedback, while highly variable across models, does have some qualitatively similar features that appear in many CMIP-class GCMs. The most salient of these is the dipole pattern in the shortwave cloud feedback (Zelinka et al., 2012b, a;Zelinka et al., 2013;Zelinka et al., 2016). The shortwave cloud feedback dipole is characterized by decreasing
cloud coverage in the subtropics (a positive feedback) and increasing cloud optical depth in the extratropics (a negative feedback) in response to warming. There is a growing consensus that the positive lobe of the dipole, where subtropical cloud fraction decreases, is a robust feature of the climate system. Both empirical analysis of observations (McCoy et al., 2017a;Clement et al., 2009;Klein et al., 1995;Myers and Norris, 2015, 2016;Norris et al., 2016) and very high resolution simulations (Blossey et al., 2013;Bretherton, 2015;Bretherton and Blossey, 2014;Bretherton et al., 2013;Rieck et al., 2012)
have substantiated the subtropical positive feedback predicted by GCMs, although it appears that traditional GCMs somewhat underpredict the decrease in subtropical cloud cover in response to warming and thus underestimate the positive feedback (Klein et al., 2017).

With the growing consensus surrounding the positive lobe of the dipole, a constraint on the negative lobe, where extratropical cloud optical depth increases with warming, has increased in importance as a significant source of uncertainty in
the global-mean shortwave cloud feedback. Evaluation of model behavior and some observations indicate that the negative lobe is related to a transition from a more ice-dominated to a more liquid-dominated state – the so called *mixed phase cloud feedback* (McCoy et al., 2017b;Ceppi et al., 2016a;Tsushima et al., 2006;Cheng et al., 2012;Naud et al., 2006;Choi et al., 2014). This transition results in an increase in small, bright liquid droplets at the expense of ice crystals and thus an increase in albedo(Zelinka et al., 2012a;McCoy et al., 2014). It is possible that this transition also decreases precipitation efficiency by
decreasing the amount of frozen hydrometeors (Field and Heymsfield, 2015;Morrison et al., 2011;Heymsfield et al., 2009), and enhancing the total condensate.



GCMs struggle to realistically simulate mixed-phase clouds. Evaluation of ice-liquid partitioning in GCMs participating in CMIP5 showed that there was a 30 K temperature range in which different models predicted an equal mixture of ice and liquid within clouds(McCoy et al., 2016). This model diversity in partitioning leads to a diversity in LWP response to warming and ultimately shortwave cloud feedback in the extratropics(Tan et al., 2016). Models that glaciate at a warmer temperature transition more ice to liquid with warming, simply because they have a large reservoir of susceptible cloud ice in the climate mean state(McCoy et al., 2015b).

The mixed-phase cloud transition mechanism is partially supported by the observationally-inferred response of extratropical clouds to warming. Several studies have substantiated that cloud optical depth responds to atmospheric and surface temperature in the midlatitudes, particularly the Southern Ocean (SO)(Ceppi et al., 2016b;Terai et al., 2016;Gordon and Klein, 2014), although there is not a strong consensus as to whether cloud optical depth increases or decrease with increasing SST when changes in atmospheric stability are considered(Terai et al., 2016). These investigations examined the variability across the midlatitudes in a non-phenominological sense, making it difficult to assign a mechanism to their diagnosed covariability between optical depth, LWP, and temperature. Overall, it remains unclear if this optical depth increase is directly related to shifts in cloud phase because it is difficult to accurately measure the phase of water in clouds and the total amount of frozen water (Jiang et al., 2012). In addition to the difficulties in measuring ice-phase cloud properties, the diversity in synoptic states in the midlatitudes further complicates this analysis. Bodas-Salcedo (2018) demonstrated that the radiative signal from increased LWP associated with phase transitions is masked by ice cloud within low pressure systems. This shows that extratropical variability in LWP needs to be considered in the context of the regime it is occurring in. This is supported by earlier studies that demonstrated that there was strong regime dependence in the bias in reflected shortwave radiation across SH extratropical cyclones (Bodas-Salcedo et al., 2014). Shifts in cyclonic regimes have been suggested as a possible explanation for negative midlatitude shortwave cloud feedback. Tselioudis and Rossow (2006) proposed that changes in cyclone frequency and surface pressure depression could increase reflected shortwave over the midlatitudes between 1.9 Wm$^{-2}$ and 4.9 Wm$^{-2}$ due to changes in intensity and frequency of cyclones in a warmed climate and the observed relationship between these properties and reflected shortwave in the current climate. Despite the complexity of midlatitude feedback processes, robust decadal increases in extratropical cloud cover (Norris et al., 2016) and liquid water path (Manaster et al., 2017) have been observed. It is reasonable to hypothesize that warming in the extratropics might be driving these trends partly via changes in cloud phase from ice to liquid. Can our understanding of the large synoptic systems that dominate the extratropics assist us in interpreting the long-term cloud property trends observed across these regions?

In this study we follow a similar technique to Tselioudis and Rossow (2006) and examine observations of midlatitude cyclones to infer a feedback. However, this is difficult to interpret in a causal sense. GCMs are used to support inferences made by examining observed covariability in the current climate. The cloud organization within midlatitude cyclone systems exists on a variety of length scales from synoptic (thousands of kilometers) to mesoscale cellular convection (kilometers). Traditional GCMs are able to capture the overall synoptic length scale, but are typically too coarse to capture the finer structures. Here, we utilize a diverse selection of GCMs with resolutions as fine as 7km to examine the impact of resolving





these features. From these simulations we hope to not only support the existence of the mechanisms we propose based on observations, but to offer guidance as to what aspects of models are important to capturing midlatitude variability and cloud feedback.

In summary, we show that the mixed-phase cloud feedback does not explain all of the observed variability or trends

in extratropical LWP within cyclones in both observations and GCMs. This is done by sorting our observations and simulations into cyclonic regimes across the extratropics. We show how clouds in cyclones have their LWP variability explained by meteorological variability and that trends in meteorological variability explain the majority of decadal trends in cyclone LWP. Similarly, changes in cyclone LWP between simulations forced with observed SST and simulations with enhanced SST can be explained by changes in moisture flux into cyclones. This work builds on earlier insight  by Kodama et al. (2014) who

utilized aquaplanet simulations to posit that a relationship between SST and WVP modulated by Clausius-Clapeyron within extratropical cyclones should lead to a negative cloud feedback, in keeping with Betts and Harshvardhan (1987). We hope that the relationships between synoptic state and cyclone cloud LWP in this work provides a clear criterion that models may be evaluated against and will reduce uncertainty related to the extratropical shortwave cloud feedback in models.

## 2 Methods

In this section we discuss the methodology used to identify the low-pressure centers of midlatitude cyclones. We compare microwave observations of cyclone properties to global model simulations ranging from CMIP5 GCMs with horizontal resolution in excess of 100 km, to convection-permitting GCMs with a resolution of approximately 7 km. The methodology used to create the unified microwave observations, cloud top phase, and the model set up for the global simulations is described in this section as well.

### 2.1 Data analysis

### 2.1.1 Regression analysis

In this work we examine observed and simulated extratropical variability in the context of linear regressions. In the cyclone compositing framework that this paper is built on we examine (i) variability of different variables between cyclones within the coordinate system of the cyclone composite (e.g. the inter-cyclone variability in some region of the composite), (ii) variability

in mean cyclone properties across many cyclones, and (iii) seasonal and regional mean variability in cyclone means (e.g. the average cyclone LWP for all cyclones in a given region). To add clarity to our analysis we will refer to a given cyclone property $X$ as $X_{CM}$ when a cyclone-wide mean is taken (where the mean is within a 2000 km radius of the low pressure center) and $X_{RM}$ when we are examining the regional mean of many different cyclones. In the case where we will investigate the spatial variability around the low-pressure center we will write $X_{ij}$ to signify the different averaging regions within the



composite. In the case of some variables only cyclone-means are defined (e.g. WCB moisture flux into the cyclone) and the 'CM' subscript is not written. A list of acronyms and subscripts is given in Table 1.

### 2.1.2 Cyclone compositing

Numerous studies have examined midlatitude variability by compositing around cyclone centers (Field et al., 2011;Field and Wood, 2007;Naud et al., 2016;Catto, 2016;Naud et al., 2017;Grandey et al., 2013). Identification of cyclone centers may be achieved by using pressure (Jung et al., 2006;Löptien et al., 2008;Hoskins and Hodges, 2002;Field et al., 2008); geopotential height (Blender and Schubert, 2000); or vorticity (Sinclair, 1994;Hoskins and Hodges, 2002;Catto et al., 2010). Here we follow the methodology described in Field and Wood (2007). Anomalies in SLP ($p_0'$) are calculated by subtracting the average of SLP from 15 days before and after at each point. Candidate grid points were found using the following criterion:

$$\frac{dp_0'}{dx}\frac{dp_0'}{dy} < 3 \cdot 10^{-5} hPa\ km^{-2} \qquad [1]$$

and

$$\frac{d^2p_0'}{dx^2} + \frac{d^2p_0'}{dy^2} > 6 \cdot 10^{-5} hPa\ km^{-2} \qquad [2]$$

where SLP<1015 hPa. As in Field and Wood (2007) SLP is averaged to 2.5° resolution, and each composite is 4000 km across. These candidate grid points are filtered to find the maximum negative anomaly within 2000km. Composited data is averaged onto an equal-area grid. The averaging grid was 18 zonal bins by 19 meridional bins. In this study we examine NH and SH cyclones. Cyclone centers must be between 30° and 80° latitude. SH cyclones are flipped in the north-south direction so that all cyclones are oriented with the pole to the top of the figure. This is done to allow easy comparison of cyclone composites in the NH and SH.

Some microwave radiometer products are unavailable over land or ice (e.g., surface wind), while others have larger uncertainty resulting from atmospheric emission signals being occasionally overwhelmed by land or ice emission (e.g., cloud liquid water). Therefore, only cyclone centers with 50% or more of the composite area located over ice-free ocean are considered valid in cyclone composites from observations or models. Model data over sea ice or land are removed from the composite to ensure parity with the observations. The number of cyclone centers identified for each GCM and the observations are shown in Fig. 1.

### 2.2 Observations

### 2.2.1 MAC-LWP

The Multi-Sensor Advanced Climatology framework used for developing monthly cloud water products (Elsaesser et al., 2017) is adapted for use here to create diurnal-cycle corrected and bias-corrected daily datasets for liquid water path (LWP, where path is the mass in an atmospheric column), 10-meter wind speed, and water vapor path (WVP).



Because passive microwave cloud liquid water retrievals must make assumptions regarding the partitioning of precipitating and non-precipitating liquid there is a systematic uncertainty in the microwave LWP data set. The cyclone LWP observations from this data set that are used in this study are the estimated non-precipitating liquid water averaged over both cloudy- and clear-sky, with the bias (largely due to the aforementioned precipitation partitioning errors) estimated to be ~0.01-

0.02 kg m$^{-2}$ for the mid-latitude regions analyzed here (Greenwald et al., 2018).

MAC-LWP uses data from multiple microwave radiometers to create a data set spanning 1988-2016. However, up until 1991 the only data source was F08 SSM/I, which therefore implies greater uncertainty in daily averages prior to 1992 (since only two satellite overpasses per day would go into such estimates).   Thus, we consider this period less reliable and only obervations onwards from 1992 are considered in this study. Because sea surface temperature and sea ice coverage are

only available through 2015 we do not examine extratropical cyclones after this period.

One possible caveat in our analysis is that the radiative signal used to retrieve LWP may partly arise from upwelling radiation due to wind roughening of the ocean surface or emission from WVP.  In such cases, LWP is biased in one direction, while wind and/or WVP may be biased in an opposite direction (Elsaesser et al., 2017). However, retrievals of WVP and wind speed have been shown to be unbiased relative to in situ observations and thus such issues are likely minimal (Mears et al.,

2001;Wentz, 2015;Trenberth et al., 2005;Meissner et al., 2001;Elsaesser et al., 2017).

### 2.2.2 MERRA2

The Modern-Era Retrospective Analysis for Research and Applications version 2(Bosilovich et al., 2015) (MERRA2) daily-mean sea level pressure (SLP) was used to locate cyclone centers in the observational record from 1992-2015 using the

algorithm described above.

### 2.2.3 AIRS Cloud-top phase partitioning

The Atmospheric Infrared Sounder (AIRS) instrument on NASA's EOS Aqua satellite provides estimates of cloud thermodynamic phase (liquid, ice, and unknown categories) (Kahn et al., 2014). The cloud phase algorithm is based on a channel selection that exploits differences in the index of refraction for liquid and ice (Nasiri and Kahn, 2008), while more

ambiguous spectral signatures are classified as unknown phase. Jin and Nasiri (2014) showed that ice cloud within the AIRS FOV is correctly identified in excess of 90% of the time when compared to estimates of thermodynamic phase from Cloud-Aerosol Lidar and Infrared Pathfinder Satellite Observation (CALIPSO; Hu et al. (2010)). Liquid phase clouds dominate subtropical stratocumulus regimes (Kahn et al., 2017) while unknown phase clouds are found most frequently in trade cumulus regimes and the cold sector of extratropical cyclones (Naud and Kahn, 2015). Observations from the ascending and descending

orbits of AIRS were averaged together to approximate a daily mean.




### 2.2.4 SST and sea ice

The Met Office Hadley Centre sea ice and sea surface temperature data set (HadISST.2.1.0.0, Titchner and Rayner (2014)) was used to provide sea ice coverage and sea surface temperature (SST) within the cyclone composite for both models and observations up until 2015. HadISST.2.1.0.0 SST and sea ice cover was also used to provide boundary conditions for the

atmosphere-only PRIMAVERA simulations described below.

### 2.3 Simulations

In this study we have assembled a broad array of GCMs to examine their midlatitude variability. Model resolutions range from quite coarse, consistent with long integrations performed as part of CMIP5, to high resolution simulations performed under the auspices of PRIMAVERA for CMIP6, UM-CASIM, ICON, and NICAM. These simulations have long integration records

and their trends may be compared to observations. Two very high-resolution simulations (nearer to 7km horizontal resolution) are also considered. Because of their demand on computational resources only short integrations are available, but they allow insight into the representation of midlatitude processes in the convective grey zone(Field et al., 2017). Simulations are described below and are listed in Table 2.

### 2.3.1 CFMIP2

We consider several models from the CMIP5 models participating in CFMIP2. These models are listed in Table 2. Atmosphere-only (AMIP) simulations using observed SST as a boundary condition are available for the period 1979-2008. In addition, simulations were performed with SST uniformly increased by 4K (AMIP+4K). The contrast between these sets of simulations will be used to investigate warming-induced changes in extratropical cyclones.

### 2.3.2 PRIMAVERA

The PRocess-based climate sIMulation: AdVances in high-resolution modelling and European climate Risk Assessment (PRIMAVERA) project is intended "To develop a new generation of advanced and well-evaluated high-resolution global climate models, capable of simulating and predicting regional climate with unprecedented fidelity, for the benefit of governments, business and society in general." Several European modelling centers have coordinated to run instances of their

CMIP6 models at increased horizontal resolution. These simulations use Easy Aerosol to unify aerosol perturbations. At the time of writing historical simulations with prescribed SST and sea ice have been completed for the models analyzed here. These simulations allow insight into whether increasing horizontal resolution impacts the ability of models to realistically represent midlatitude variability. High resolution models are labeled HR and low resolution is labeled LR, with the exception of HadGEM3, which has three resolutions. PRIMAVERA simulations are performed under the HighResMIP protocols

outlined by the climate model intercomparison project panel.



### 2.3.2.1 EC-Earth

The EC-Earth model used for HighResMIP/PRIMAVERA is part of the EC-Earth3-familiy. EC-Earth 3 is a successor of the version 2.3 used for CMIP5 (Hazeleger et al., 2012). The version used in HighResMIP is EC-Earth3.2.P. Compared to version 2.3, EC-Earth3.2.P includes updated versions of its atmospheric and oceanic model components, as well as a higher horizontal

and vertical resolution in the atmosphere.

The atmospheric component of EC-Earth is the Integrated Forecast System (IFS) of the European Centre for Medium Range Weather Forecasts (ECMWF). Based on cycle 36r4 of IFS, it is used at T255 and at T511 resolution for the standard and high resolution simulation in HighResMIP, respectively. It uses a reduced Gauss-grid with 91 vertical levels. The nominal resolution is about 100km x 100km in standard resolution and 50 x 50 km in high resolution.

The ocean component is the Nucleus for European Modelling of the Ocean (NEMO, (Madec, 2008)). It uses a tri-polar grid with poles over northern North America, Siberia and Antarctica with a resolution of about 1 degree (the so-called ORCA1-configuration) and 75 vertical levels (compared to 42 levels in the CMIP5 model version) in the standard resolution. In high resolution, the ORCA025 configuration is used with a resolution of about 0.25 degree.

The ocean model version is based on NEMO version 3.6 and includes the Louvain la Neuve sea-ice model version 3 (LIM3,

(Vancoppenolle et al., 2012)), which is a dynamic-thermodynamic sea-ice model with five ice thickness categories.

The atmosphere and ocean/sea ice parts are coupled through the OASIS (Ocean, Atmosphere, Sea Ice, Soil) coupler.

The high-resolution configurations (T511 atmosphere and ORCA025 ocean, coupled or stand-alone) have been newly developed for EC-Earth 3. The high-resolution NEMO configuration is based on a set-up developed by the ShaCoNEMO collaboration and adapted to the specific atmosphere coupling used in EC-Earth. Particularly, the remapping of runoff from

the atmospheric grid points to runoff areas on the ocean grid has been re-implemented to be independent of the grid resolution. This is done by introducing an auxiliary model component and relying on the interpolation routines provided by the OASIS coupler. In a similar manner, forcing data for the atmosphere is passed through a separate model component, which allows use of the same forcing data set for different EC-Earth configurations.

A full description of EC-Earth3.2.P and its ability to simulate the climate can be found in Haarsma (2018).

### 2.3.2.2 HadGEM3

HadGEM3-GC3.1 is described in Williams et al. (2018). The atmospheric only simulations used in this paper comprises component configurations Global Atmosphere 7.1 (GA7.1), Easy Aerosol, and JULES Global Land 7.1 (GL7.1) described in Walters et al. (2017). GA7.1 dynamical core ENDGame uses a semi-implicit semi-Lagrangian formulation to solve the non-hydrostatic, fully-compressible deep-atmosphere equations of motion (Wood et al., 2014). The microphysics used is based on

Wilson and Ballard (1999), with extensive modifications described in more details in Walters et al. (2017). The parametrisation used is the prognostic cloud fraction and prognostic condensate (PC2) scheme (Wilson et al., 2008a;Wilson et al., 2008b) along with the cloud erosion parametrisation described by Morcrette (2012) and critical relative humidity parametrisation



described in Van Weverberg et al. (2016). The model uses 85 vertical levels with 50 levels below 18 km and 35 levels above this, and a fixed model lid 85 km above sea level. Three different horizontal resolutions of the regular lat-lon grid are used in this study: N96, N216 and N512, which correspond respectively to a grid cell size of 135km, 60km and 25km at 50ºN, and are referred to as LM, MM and HM in the rest of the paper. The UM uses a mass flux convection scheme based on Gregory and

Rowntree (1990) with various extensions to include down-draughts (Gregory and Allen, 1991) and convective momentum transport (CMT).

### 2.3.2.3 CNRM-CM6

The atmospheric only simulations analysed in this study are based on the atmosphere-land component of CNRM-CM6 which consists in the atmospheric model ARPEGE-Climat version 6.3, fully described in (Roehrig, 2018), and the SURFEX v8 land

surface scheme(Decharme, 2018). The ARPEGE-Climat dynamical core is derived from IFS cycle 37t1. The model is operated with a T127 and a T359 truncation, the associated horizontal resolution being 120 km and 50 km for the LR and HR versions respectively. In both versions there are 91 vertical levels in the atmosphere. Compared to CNRM-CM5, the atmospheric physics has been largely revisited. In particular, convection scheme, microphysics scheme and turbulent scheme have been updated. The convection scheme (Guérémy, 2011;Piriou et al., 2007) provides a consistent, continuous, and prognostic

treatment of convection from dry thermals to deep precipitating events. The microphysics scheme is derived from Lopez (2002) and takes into account autoconversion, sedimendation, ice-melting, precipitation evaporation and collection. The turbulence scheme represents the TKE with a 1.5-order scheme prognostic equation according to Cuxart et al. (2000). Surface drag over oceans is capped in CNRM-CM6 (see Soloviev et al. (2014) for general discussion). The calculations of exchange coefficients over ocean are based on an updated version of the Exchange Coefficients from Unified Multi-campaigns Estimates (Belamari,

2005) scheme.

### 2.3.3 NICAM

NICAM (Satoh et al., 2008;Satoh et al., 2014;Tomita and Satoh, 2004) is a non-hydrostatic atmospheric model with the icosahedral grid system. Here, climate simulation output from 14 km mesh NICAM(Kodama et al., 2015) is used for an analysis. Horizontal resolution is approximately 14 km, and 38 vertical levels are configured up to around 40 km. Instead of using

convection and large-scale condensation schemes, a single moment bulk cloud microphysics scheme (Tomita, 2008) is used, in which rain, snow, and graupel as well as water vapor, cloud water, and cloud ice are treated as prognostic variables. SST is not fixed but nudged toward its monthly-mean historical distributions (Kodama et al., 2015).

### 2.3.4 ICON

The experiment using the ICOsahedral Non-hydrostatic (ICON) atmospheric model applied here uses a non-hydrostatic dynamical core, like NICAM, on the icosahedral grid (Zängl et al., 2015). The 1-year run was conducted as part of a



development towards kilometer-scale global simulations, and as such should be considered preliminary. The grid applied here has an equivalent grid-spacing of 10 km, and in the vertical 70 levels are applied with a top around 30 km. The atmospheric physics parameterizations are from the ICON-ESM (Giorgetta et al., 2018) typically applied at much lower resolutions, but here adapted to convective cloud-permitting scales. This includes turning off all moist convective parameterizations, shallow-

mid- and deep convection, as well as disabling all sub-grid scale gravity wave parameterizations and changing certain tuning parameters. These changes were to set the critical relative humidity for cloud formation everywhere to unity, setting the sub-grid scale cloud inhomogeneity factors to unity, and setting the turbulence parameterization near-neutral turbulent Prandtl number to 0.7. The input data used in this simulation was later found to contain a series of problems, however deemed irrelevant for the purposes of this study.

**2.3.5 UM-CASIM**

The simulations presented here are described fully in McCoy et al. (2018b) – the following description is adapted in brief below. Simulations were performed in the MetOffice Unified Model (UM) vn10.3 based on GA6 (Walters et al., 2017) in a convection-permitting setting in aquaplanet mode (no continents or sea ice). The model was run at 0.088°x0.059° and neither convection parametrization nor cloud scheme were used. Simulations lasted for 15 days and were run with 70 vertical

levels.  The Cloud-AeroSol Interacting Microphysics (CASIM) two-moment microphysics scheme (Hill et al., 2015;Shipway and Hill, 2012;Grosvenor et al., 2017;Miltenberger et al., 2018) was used and is described in Shipway and Hill (2012). The warm rain processes in CASIM is compared to other microphysics schemes in Hill et al. (2015). The rain autoconversion and accretion rates parameterization used in CASIM are described in Khairoutdinov and Kogan (2000).  Because these simulations are run in GA6 with CASIM microphysics they should not be directly compared to the HadGEM3 simulations in

PRIMAVERA described above.

Sea surface temperature (SST) was held fixed in the simulations and the atmosphere was allowed to spin up for a week at low resolution and then for another week at high resolution. The SST profile used in the aquaplanet was derived from a 20-year climatology run from the UM in standard climate model configuration. The January SST was reflected north-south. The orginal and reflected SST were averaged together. The resulting SST was zonally averaged to produce a symmetrical SST.

Aerosol concentration is constant in the simulations. The aerosol profile was 100 cm$^{-3}$ in the accumulation mode at the surface up until 5km and then exponentially decreased after 5km with an e-folding of 1 km. Aerosol-cloud interactions were parameterized using a simple Twomey-type parameterization of cloud droplet number concentration (CDNC) (Rogers and Yau, 1989) $CDNC = 0.5 N_{acc} \text{w}^{0.25}$ with $N_{acc}$ being accumulation mode aerosol number concentration and w being updraft velocity limited such that at w=16m/s CDNC= $N_{acc}$. The aerosol forcing in these simulations is highly idealized and is not

intended to represent any sort of variation in aerosol properties in the same way as Easy Aerosol. The vertical velocity was set to have a minimum value of 0.1m/s. Ice number was controlled using a simple temperature-dependent relationship (Cooper, 1986).  Because only two weeks of simulations were available for the UM-CASIM runs contours of SLP (as opposed to




anomalies in SLP relative to the monthly-mean) were used to identify candidate cyclone centers as described in McCoy et al. (2018b).

## 3 Results

### 3.1 Precipitation and WCB moisture flux

5   The majority of moisture ingested into extratropical cyclones is imported along the warm conveyor belt (WCB) (Eckhardt et al., 2004;Field and Wood, 2007). The WCB moisture flux is defined as

$$WCB = k \cdot WVP_{CM} \cdot WS_{10mCM} \tag{3}$$

Where $WVP_{CM}$ is the cyclone-mean water vapor path in kg/m$^2$ ; $WS_{10mCM}$ is the cyclone-mean wind speed at 10 meters in m/s; and $k$ is a constant parameterizing the width of the WCB as defined in Field and Wood (2007) and is calculated by

10 linear regression of the precipitation rate on $WVP_{CM} \cdot WS_{10mCM}$ . Cyclone means are within 2000 km of the cyclone center. We note that the $k$ in Field and Wood (2007) was based on AMSR-E data. It is likely that AMSR-E misses around 50% of the precipitation in extratropical cyclones (Naud et al., 2018;Field et al., 2011). In this study we will use $k = 2.66 \times 10^{-7} \, m^{-1}$, consistent with AMSR-E.

   Although the moisture imported along the WCB may condense and form clouds within the cyclone, in order to

15 maintain water mass balance in extratropical cyclones the moisture flux into a cyclone must match the precipitation out of the cyclone over a 2000 km radius. We examine whether this holds in the models listed in Table 2 (Fig. 2). All the models show a very similar relationship between moisture flux into cyclones and precipitation rate averaged across the cyclone. Model values of the WCB width parameter ($k$) (Eq. 3) range from $2.41 \times 10^{-7} \, m^{-1}$ to $4.11 \times 10^{-7} \, m^{-1}$ and are generally higher than the $k$ trained on AMSR-E data(Field and Wood, 2007). It is interesting to note that the $k$ value does not appear to depend

20 on model resolution and the lowest and highest $k$'s come from the high resolution simulations in ICON and UM-CASIM, respectively. However, this range in $k$ is within the observational uncertainty in precipitation rate(Field et al., 2011). Naud et al. (2018) examined observed precipitation rate in extratropical cyclone and found that the mean extratropical cyclone precipitation rate differed substantially depending on whether a microwave radiometer (0.08 mm/hr, AMSR-E) or radar (0.17 mm/hr, Cloudsat) was used to measure precipitation rate. If we rescale the AMSR-E precipitation rates so that the cyclone-

25 mean precipitation rate is consistent with radar measurements, $k$ should be $5.67 \times 10^{-7} \, m^{-1}$. Overall, the GCM cyclone precipitation flux that is predicted by the simple model of WCB moisture flux and the $k$ inferred from the GCMs is well within the observational uncertainty.

   Given the non-linear nature of Eq. 3, and the societal and economic importance of precipitation rates over the heavily populated NH midlatitudes, WCB moisture flux provides a useful constraint on precipitation- both in the climate mean-state

30 and in projected changes in rain rate via dynamical alterations (in wind speed) and Clausius-Clapeyron driven changes (in WVP). We compare the distributions of WCB moisture flux, $WS_{10mCM}$, and $WVP_{CM}$ in models and observations. The mean WCB moisture flux in the GCMs considered in this study is generally lower than the observations (Fig. S1 ab), with model





biases ranging from -1.16 mm/day to -0.31 mm/day in the SH and from -0.79 mm/day to +0.24 mm/day in the NH. This bias appears to be linked to low 10-meter wind speed in cyclones in models (Fig. S1 cd) as GCM $WVP_{CM}$ is near to the observed distribution (Fig. S1 ef). One possibility is that this issue is related to excessive surface drag over oceans, which is a known issue in modelling tropical cyclones(Donelan et al., 2004;Soloviev et al., 2014). Anecdotally, the CNRM-CM6 LR and HR

GCMs cap surface drag and are the only two GCMs whose mean wind speed is greater than or equal to the observed wind speed. Based on this we suggested sensitivity tests in GCMs to the capping of surface drag as a step toward a realistic representation of midlatitude precipitation rates. If this is the cause of lower surface wind speed in cyclones then it means that midlatitude cyclones have been systematically under-estimating precipitation through decreased flux of moisture into the cyclone. It is also possible that the low $WS_{10mCM}$ in some of the GCMs reflects deficient horizontal resolution (Strachan et al.,

2013). The most biased cyclone-mean $WS_{10m}$ speeds are the IPSL-CM5 and CNMR-CM5 models, which have relatively low horizontal resolutions, but this may be concidental as there does not appear to be a systematic trend in as horizontal resolution increases in different instances of the same model within the PRIMAVERA GCMs.

**3.2 LWP and WCB moisture flux**

As shown in section 3.1, precipitation within midlatitude cyclones is predicted by WCB moisture flux. This means that cyclones are in an approximate steady state because the flux of moisture into the cyclone is matched by the flux of precipitation out of the cyclone. In this study we will examine how this steady state varies across GCMs and observations. In particular, we will examine the transition of water vapor to precipitation through its intermediary state suspended in cloud droplets.

If extratropical cyclones are in steady-state, then we expect that an increased moisture flux should enhance cyclone LWP, providing that precipitation processes are dominated by the warm rain process. This is because a higher LWP is needed to generate a higher rain rate (Wood et al., 2009;Hill et al., 2015). We will elaborate on this assumption shortly. Enhanced cyclone LWP should either increase in-cloud LWP or increase cloud coverage. Both effects should translate to an enhancement in cyclone albedo (at a fixed solar zenith angle, see the discussion in McCoy et al. (2018b)). This makes understanding the

efficiency with which extratropical cyclones can convert moisture flux to precipitation via cloud water key in understanding variability in extratropical albedo. We note that in this study we utilize microwave observations of LWP, which are the average of cloudy and clear regions, so increases in either *in-cloud* LWP or cloud coverage should translate to an increase in microwave-observed LWP. Similarly, the GCM LWP is the average of clear and cloudy regions.

Extratropical cyclone LWP represents a key variable in determining extratropical albedo, but does it scale with WCB

moisture flux? A linkage between moisture flux into an extratropical cyclone and the total column liquid in the cyclone has been demonstrated previously in McCoy et al. (2018b). A caveat to this is that in McCoy et al. (2018b) total liquid water path (TLWP, precipitating and non-precipitating liquid) was examined. Here, we examine the fraction of the TLWP which is suspended in clouds (referred to as LWP, here). Does $LWP_{CM}$ increase with WCB moisture flux in the same way that $TLWP_{CM}$





does? The efficiency with which extratropical cyclones can shift vapor to rain determines the relation between WCB moisture flux and $LWP_{CM}$. In the limiting case this efficiency might increase sufficiently rapidly with moisture flux that $LWP_{CM}$ would not increase in step with WCB moisture flux (all additional liquid becomes rain). Because we cannot directly observe how extratropical cyclones partition precipitating and non-precipitating liquid (see methods section), we cannot directly evaluate

how precipitation efficiency scales with WCB moisture flux.  This represents an uncertainty in our analysis.  However, we can evaluate extratropical cyclone rain and cloud partitioning in high-resolution simulations as a check on the caclulations used in the MAC-LWP dataset. $LWP_{CM}$, $TLWP_{CM}$ (rain+cloud liquid), and WCB moisture flux are calculated for extratropical cyclones observed using MAC-LWP and as simulated by UM-CASIM (see Table 2). The parameterization that partitions cloud and rain water paths in the MAC-LWP observations results in a decreases the fraction of total liquid path that is in clouds

($LWP_{CM}/TLWP_{CM}$) as WCB moisture flux decreases by -0.075 day/mm (see the slope of the line in Fig. S2). Comparison to UM-CASIM simulations shows a similar decrease in the fraction of liquid water that is suspended in clouds (-0.087 day/mm, Fig. S2).  Ultimately, the partitioning of rain and cloud water in MAC-LWP, and the microphysics scheme in UM-CASIM both lead to an increase in LWP with increasing WCB moisture flux. Conceptually, we should expect this based on the warm rain process. To reiterate, a greater LWP is required to yield a larger precipitation(Wood et al., 2009;Hill et al., 2015), which

is in turn needed to match the moisture flux into the cyclone.

       With the caveat in mind that we must infer the partitioning of liquid between rain and cloud, we evaluate the ability of a wide array of GCMs to simulate the response of cyclone-mean LWP ($LWP_{CM}$) to WCB moisture flux. We compare the WCB moisture flux dependence of $LWP_{CM}$ in the models listed in Table 2 and observations from MAC-LWP (Fig. 3). Increasing WCB moisture flux increases $LWP_{CM}$ in both observations and models. While the high-resolution models (<100km

horizontal resolution) have a slope of the WCB-$LWP_{CM}$ relationship that is in keeping with the observed slope, they tend to have too low a $LWP_{CM}$ for a given WCB moisture flux. However, if the maximum bias in observed $LWP_{CM}$ of 0.03 kg/m$^2$ is assumed based on an estimated range 0.01-0.02 kg/m$^2$ (Greenwald et al., 2018), then many of these models are in the possible observational range. It is also reasonable to suspect that models that only generate clouds when the entire grid box is saturated (e.g. there is no convection parameterization or cloud scheme) will under-estimate cloudiness.

It is suggestive that the lower-resolution CFMIP2 models tend to have a much wider diversity in slopes than the higher resolution PRIMAVERA models, UM-CASIM, NICAM, and ICON. This may reflect parametric uncertainty in the representation of convection. UM-CASIM, NICAM, and ICON do not parameterize convection and have extremely similar relationships between WCB moisture flux and LWP. Based on this we suggest that the relationship between moisture flux and LWP may offer a possible evaluation tool for the realism of convection within GCMs. However, this may also just be chance

related to the selection of models presented here as the low-resolution HadGEM3-GC31-LM has a reasonably close behavior to the higher resolution instances of that model (HadGEM3-GC31-MM, and HadGEM3-GC31-HM). Overall, the constraint provided by the WCB-$LWP_{CM}$ relationship shown here provides a useful tool for GCMs to evaluate their climate mean-state behavior in the extratropics.



As discussed in the paragraph above, the partitioning between rain and cloud liquid shifts toward rain at higher moisture fluxes (Fig. S2). This leads to the asymptotic nature of the curves shown in Fig. 3. Presumably this reflects differences in the way that precipitation is treated in the different GCMs with (for example) autoconversion being stronger in some models leading to a more pronounced flattening of the curve at higher LWP as precipitation becomes more efficient. In this work we will treat the asymptotic behavior of the WCB-LWP$_{CM}$ curve as a second order effect for the sake of simplicity in our analysis. However, we note that this behavior does provide a useful evaluation of the precipitation processes in a given model and more in-depth examination of this feature is reserved for a future study.

### 3.3 Long-term variability in observed cloud properties

### 3.3.1 Monthly-mean regional variability in extratropical cyclone properties

The moisture flux into extratropical cyclones plays a dominant role in determining their LWP and, ultimately, precipitation rate. How does this mechanism influence the cloud feedback in the midlatitudes? In keeping with earlier studies(Myers and Norris, 2016) we examine observed anomalous variability from 1992-2015 to infer the cloud feedback in these regions within cyclones. The Indian, Pacific, and Atlantic oceans between 30°S-80°S; and Atlantic, and Pacific oceans poleward of 30°-80°N are each examined individually (Fig. 4). In this section we discuss cyclone-means in the context of the monthly-means across all the cyclones in each region. For each region the monthly-mean anomaly relative to the climatology is calculated. Variables averaged to regional means are denoted RM (see Table 1 for a list of acronyms and subscripts).

Before we discuss analysis of anomalous variability in extratropical cyclones, we want to note that in McCoy et al. (2018b) TLWP$_{CM}$ (rain and cloud) was fit using the form $TLWP_{CM} = a \cdot WCB^b CDNC_{SW}^c + d$, where $CDNC_{SW}$ was the average cloud droplet number concentration (CDNC) within the southwest quadrant (in poleward coordinates, alternatively equator-westward) and WCB was WCB moisture flux. As shown in Fig. 3, there is a power-law relationship between LWP$_{CM}$ and WCB moisture flux across observations and models. Here we will linearly relate monthly-mean anomalies in LWP$_{RM}$ to regional- and monthly-mean anomalies in WCB moisture flux. A linear relation is used in this case because the anomalous variability is relatively small (compared to the overall variability) and the relation between monthly-mean anomalies in LWP$_{RM}$ and anomalies in regional- and monthly-mean WCB moisture flux is approximately linear.

Fig. 5 shows the relation between the regional- and monthly mean of of various cyclone properties. For example, we may ask if across a given ocean basin in a given month the LWP within cyclones is higher when WCB moisture flux into cyclones is higher (Fig. 5a). For each ocean basin (Fig. 4) the average of the LWP$_{CM}$ for all cyclones for each month is taken (LWP$_{RM}$). The climatological LWP$_{RM}$ is subtracted for each month to yield anomalies. The same procedure is repeated for WCB moisture flux. The relation between anomalies in LWP$_{RM}$ and WCB moisture flux anomalies is shown in Fig. 5a. This allows us to examine the relation of various predictors across the population of cyclones within a given basin.

Anomalous variability in LWP$_{RM}$ in the SH oceans correlated with variability in WCB moisture flux (28-42%, Fig. 5a). The South Pacific region has 42% of monthly-mean LWP$_{RM}$ anomalies explained by moisture flux anomalies, the South



Atlantic and Indian Oceans have approximately 30% of their monthly anomalies in LWP$_{RM}$ explained by moisture flux. Overall, the slope of the relation between anomalies in monthly-mean extratropical cyclone LWP$_{RM}$ and WCB moisture flux monthly means are quite similar across these regions. As discussed above, the methodology here assumes linearity in the change in LWP$_{RM}$ in response to anomalies in regional and monthly-mean WCB moisture flux. One possibility to explain the

range of $R^2$ across basins is that larger ranges in WCB moisture flux may lead to non-linearity in the relationship and thus a lower $R^2$ for the fit between anomalies in LWP$_{RM}$ and WCB. However, the range of WCB anomalies is only slightly smaller in the SH (Fig. 5bc).

It is also interesting to speculate on the potential effect of variability in cloud condensation nuclei (CCN) on the relationship between WCB moisture flux and LWP across basins. Enhanced CCN enhances cloud droplet number

concentration (CDNC) and inhibits the warm rain process in cyclones and enhances LWP at a given WCB moisture flux (McCoy et al., 2018b). Thus, anomalous inter-annual variability in CCN would diminish the fraction of the variance explained by WCB moisture flux alone (e.g. decrease $R^2$). As shown in previous studies, the primary source of variability of CCN in the Southern Ocean is biogenic sulfate (Ayers and Cainey, 2007;Ayers and Gras, 1991;Meskhidze and Nenes, 2006, 2010;Charlson et al., 1987). The Pacific basin of the Southern Ocean is less biologically productive and does not have the

intense phytoplankton blooms present in the Atlantic basin(McCoy et al., 2015a). It is possible that the low summer-summer variability in biogenic CCN in the Pacific leads to a greater fraction of the 1992-2015 anomalous monthly variability being explained by meteorological drivers, while the intense, but intermittent blooms and accompanying CCN in the South Atlantic and Indian Oceans may diminish the fraction of total variability contributed by meteorology.

It is interesting to contrast the SH ocean basins with the northern Atlantic and Pacific (Fig. 5a). Only 30% of North

Pacific and North Atlantic monthly-mean LWP$_{RM}$ variability is explained by WCB moisture flux. However, the best fit line in the NH oceans is similar to the best fit in the SH oceans - indicating that the WCB-moisture flux mechanisms are likely to be at work controlling inter-annual variability, but is not as relevant in relation to observed anomalous variability in monthly-mean LWP$_{RM}$ in the last two decades. It may be that strong year-year variability in anthropogenic sulfate, and thus CDNC, in the NH ocean basins(McCoy et al., 2018a) reduces the fraction of anomalous variability that is explained by moisture flux

alone. In fact, pollution control measures in both East Asia and North America have led to a steady downward trend in CDNC across their midlatitude outflow regions in recent years (McCoy et al., 2018a;Krotkov et al., 2016), following a period of enhancing CDNC from 1980-2005 off the coast of China (Bennartz et al., 2011). The effects on LWP from the trend in CDNC over the period 2005-2015 off the coast of East Asia and from 1992-onwards off the east coast of North America should oppose the effects from the trend in WCB moisture flux. This is because decreased CDNC should decrease LWP for a given WCB

moisture flux, and as discussed below, moisture flux will be enhanced by Clausius-Clapeyron-driven enhancement in WVP as the oceans and atmosphere warm. Enhancement in WCB moisture flux should enhance cyclone LWP. Thus, the meteorologically-driven trend and microphysically-driven trends in these regions are likely to be opposed, at least in the last decade. Finally, we may speculate that land-ocean interactions in the NH affects cyclone properties via mechanisms such as cold-air outbreaks, which may in turn affect LWP (McCoy et al., 2017c).





We should also note that cyclones at low latitudes are likely to be transitioning from tropical to extratropical cyclones. Tropical cyclones differ in their meteorological drivers and the WCB moisture flux mechanism is not relevant to their development (Emanuel, 2003). If only cyclones centered between 35°-80° latitude are considered, the variance explained in $LWP_{RM}$ by WCB moisture flux increases from 30-42% to 36%-46% (Fig. S3). However, the correlation between anomalies in WCB moisture flux and $LWP_{RM}$ are significant at 95% confidence, regardless of the latitude range considered.

A large fraction of SH ocean anomalies in monthly-mean $LWP_{RM}$ and, ultimately, 32% of anomalous monthly-mean $LWP_{RM}$ variability across all basins may be explained by WCB moisture flux (weighting each basin equally in the linear regression). What in turn explains moisture flux variability? As shown in Eq. 3, WCB moisture flux is the product of WVP and wind speed. We will discuss the contributions of each of these terms below.

Monthly-mean cyclone $WS_{10mRM}$, which is a proxy for the input rate of moisture into the cyclone, enhances as cyclones move poleward in both hemispheres (Fig. 5b and Fig. S4). In the Southern Oceans and North Atlantic 18-55% of anomalous monthly variability in wind speed is linearly related to cyclone poleward latitude- in the North Pacific only 18% of anomalous variability in wind speed may be explained by mean cyclone latitude. Examination of cyclone-mean wind speed as a function of latitude shows agreement in models and observations. In models and observations cyclone-mean wind speed increases toward 60°, and then decreases poleward of 60° in both hemispheres (Fig. S4). Overall, the explained variance in anomalous monthly- and regional-mean wind speed is 41% across all basins (weighting all ocean basins equally). This reflects the genesis and development of an extratropical cyclone. The genesis of extratropical cyclones occurs toward the tropics and then over their life cycle cyclones move toward the pole. During this life cycle they intensify, leading to enhancement in near-surface wind speed (Tamarin and Kaspi, 2017;Beare, 2007;Bengtsson et al., 2009). Two important questions stand out in regards to our analysis: Will the genensis region of extratropical cyclones shift in a warmed climate? How will extratropical cyclones develop in a warmed climate?

There is a general consensus that storm tracks will shift toward the poles as the climate warms(Barnes and Polvani, 2013;Yin, 2005;Lorenz and DeWeaver, 2007;Bender et al., 2011b), but the mechanism that prompts this poleward movement remains unclear (Shaw et al., 2016). As they shift poleward storm tracks intensify(Lorenz and DeWeaver, 2007;Yin, 2005). Alterations in tropopause height have been suggested as the mechanism underlying this change(Lorenz and DeWeaver, 2007). The relationship between storm track behavior and cyclone behavior adds additional complexity. Ulbrich et al. (2009) provides a thorough review of studies investigating changes in extratropical cyclone intensity, placement, and population in warmed climates. Simulations with green house gas warming generally show decreased frequency of midlatitude cyclones, but increases in cyclone intensity (Lambert and Fyfe, 2006;Bengtsson et al., 2006;Geng and Sugi, 2003). Changes in extratropical cyclone life cycle further complicates predicting changes in cyclone wind speed in a warming world. Utilizing a single highly-idealized GCM Tamarin and Kaspi (2017) demonstrate that extratropical cyclones not only shift poleward, but take longer in their development in a warming world with peak intensity being reached after a greater poleward propogation. Further investigation within the CMIP5 models by Tamarin-Brodsky and Kaspi (2017) demonstrated that this is a robust feature of GCMs and is not limited to highly-idealized models. Overall, the complexity of changes in the life cycle, frequency, and



intensity of extratropical cyclones under warming makes it difficult to  say confidently how their vorticity and surface wind speed will change. It is interesting to speculate that substantial changes in cyclone voriticity might have occurred in past cold climates such as the Maunder minimum (Raible et al., 2007). We reserve further evaluation of changes in cyclone wind speed in high-resolution simulations for integration of the warming simulations as part of PRIMAVERA. We will discuss the climate

response within CFMIP2 GCMs where the prescribed SST is enhanced by 4 K in the following section.

Monthly-mean variability in extratropical cyclone $\ln(WVP)_{RM}$ is explained by $SST_{RM}$ with 76% of anomalous monthly mean variability in $WVP_{RM}$ linked to $\ln(SST)_{RM}$. The explained anomalous variance in individual ocean basins is greater than 70%, with the exception of the North Pacific, where it is 65%. This linkage between anomalies in cyclone-mean SST and anomalies in WVP via Clausius-Clapeyron has been shown previously in Field et al. (2008).

We propose that Southern Ocean cloud feedbacks in cyclonic systems are not only related to the so-called mixed-phase cloud feedback, but are contributed to by changes in WVP and wind speed. Because increasing SST increases WVP via Clausius-Clapeyron it is easy to conflate it with ice to liquid transitions driven by SST increases. We will examine this hypothesis in the following sections.

### 3.3.2 Global model-observation comparisons of extratropical cyclone behavior

As shown above, observed extratropical cyclone $LWP_{CM}$ depends strongly on WCB moisture flux. This translates to anomalous regional- and monthly-mean variability in WCB moisture flux strongly covarying with regional- and monthly-mean anomalous variability in extratropical cyclone $LWP_{RM}$. As we saw in Fig. 3, climate model extratropical cyclone LWP depends on WCB moisture flux, but models do not agree on how sensitive cyclone $LWP_{CM}$ is to moisture flux. In this section we examine how GCM regional- and monthly-mean anomalous variability in extratropical cyclone properties compares to observations.

First, we examine the ability of models to reproduce the WCB moisture flux-$LWP_{RM}$ relation observed in the SH. As in Fig. 5a, the slope of the best fit line between monthly-mean anomalies in cyclone $LWP_{RM}$ and WCB moisture flux is computed in each SH ocean basin, and is summarized in Fig. 6a.  The 95% confidence on the best fit line is also shown. All the models and the observations have a non-zero slope at 95% confidence. The slope of the WCB moisture flux LWP relationship in IPSL-CM5B-LR, and CNRM-CM5 models are more than twice the slope inferred from observations, while the

CNRM-CM6 and HadGEM3 models have around half the observed slope. NICAM, HadGEM2, IPSL-CM5A-LR, and the EC-Earth models compare favorably to the observations. Evaluation of model variability shows that all models have over 20% of their $LWP_{RM}$ variability explained by WCB in the SH, with some models able to explain up to 70% of their anomalous monthly- and regional-mean variability using WCB moisture flux (Fig. S5).

Next, we investigate the relation between mean absolute (poleward) cyclone latitude and $WS_{10m}$ in cyclones. All

models have a correlation between anomalous monthly-mean poleward latitude and $WS_{10mRM}$ at 95% confidence (Fig. 6b), in keeping with the agreement in the latitudinal dependence of $WS_{10mCM}$ shown in Fig. S4. GCMs tend to have a higher sensitivity to mean cyclone latitude than the observations, but are in good overall agreement. This supports the idea that the models presented here have a fairly consistent representation of the cyclone life cycle in the current climate.



Finally, we examine the relation between $\ln(WVP)_{RM}$ and $SST_{RM}$ in the GCMs. The relation between SST and column water vapor in the models and in the observations are quite similar (Fig. 6c), indicating that all the models are able to somewhat accurately reproduce the response in WVP associated with Clausius-Clapeyron and warming.

### 3.3.3 Decadal trends in extratropical cyclone properties

5       We have discussed how monthly anomalies in $LWP_{RM}$ may be explained by moisture flux. This supports the idea that the shortwave cloud feedback in the Southern Ocean (SO) may be partially driven by changes in meteorology and not only by ice to liquid transitions. However, the robust decadal trend in zonal-mean LWP in the SO observed by Manaster et al. (2017) still needs to be discussed. Analysis of the decadal trend in observational record to infer climate response would interpret this as confirmation for a negative shortwave cloud feedback in the SO. However, it is unclear what has caused this change in
LWP. We will now examine the trend in LWP during the period 1992-2015 in the context of trends in moisture flux.

      First, in this study we are pursuing a regime-oriented approach to understanding extratropical variability. Do the zonal-mean trends in Manaster et al. (2017) agree with the trends in extratropical cyclone behavior? Because Manaster et al. (2017) investigated trends in the latitude band 44.5°S-59.5°S we subset our data record to only consider cyclones centered in this latitude band. Trends in Southern Ocean regional-mean cyclone $LWP_{RM}$ and zonal-mean LWP as calculated by Manaster
et al. (2017) over the last two decades are similar (Fig. 7ab, 2.40±0.58 g m$^{-2}$ decade$^{-1}$ within extratropical cyclones versus 1.8±0.8g m$^{-2}$ decade$^{-1}$ in the zonal-mean(Manaster et al., 2017), where uncertainty is the 95% confidence interval).

      Given that cyclones cover approximately half the Southern Ocean (Bodas-Salcedo et al., 2014), this in-cyclone trend can account for a good portion of the overall zonal-mean signal. How does this trend partition into components related to meteorology (as characterized by WCB moisture flux) and thermodynamics (as characterized by SST)? We investigate this
utilizing a simple regression model

$$LWP_{CM} = a \cdot WCB + b \cdot SST_{CM} + c \qquad [4]$$

This regression model is trained on the population of SH cyclone-means from 1992-2015 and centered in the latitude band considered in Manaster et al. (2017). It does not capture some of the power-law behavior in the WCB moisture flux-LWP relationship, but it simplifies the interpretation of the anomalous variability. The regression model trained on the observational
record has coefficients

$$LWP_{CM} = (28.71 \pm 0.42) \cdot WCB - (0.29 \pm 0.1) \cdot SST_{CM} + 3.70 \pm 1.21, n = 18842, R^2 = 0.53 \qquad [5]$$

 The trend in regional- and cyclone-mean LWP predicted based on this regression model and changes in SST are relatively slight (Fig. 7bc)- most of the long-term trend in $LWP_{CM}$ averaged across the Southern Ocean can be explained by changes in WCB moisture flux (Fig. 7bd). In turn, most of the increase in WCB moisture flux may be explained by steadily increasing
cyclone WVP, driven by enhanced SST (Fig. S6). The steady enhancement of SST across the Southern Ocean is driven by anthropogenic forcing (Liu and Curry, 2010).

      It is important to state the caveat that WCB and $SST_{CM}$ are fairly colinear (r=0.84 over the 1992-2015 period in both hemispheres). However, despite variance being shared between SST and WCB moisture flux, the coefficient relating WCB



moisture flux and LWP in Eq.5 is relatively insensitive to whether or not SST is included as a predictor. If WCB moisture flux is used as the only predictor, then the coefficient relating WCB moisture flux to LWP changes to 28.13±0.40 g m$^{-2}$ day mm$^{-1}$. If only SST is used as predictor, then the coefficient relating SST and LWP changes sign (+2.90±0.12 g m$^{-2}$ K$^{-1}$). Based on this we feel that the relationship between WCB moisture flux and LWP as inferred by multiple linear regression is fairly

robust, despite sharing significant variability with SST.

It appears that once WCB moisture flux is accounted for relatively little room is left for an effect related to phase changes. SST may not act as a good predictor of the ice-to-liquid transition, but the residual trend unrelated to SST trends or WCB trends is small. That is to say, we cannot take the trends in the observational record of cyclone LWP as a sign of a strong mixed-phase cloud feedback because most of the trend is explained by changes in synoptic state. We will support the

observationally-inferred importance of WCB moisture flux in a warming climate using GCM simulations with artificially-enhanced SST below.

Of course this only examines variability within extratropical cyclones. One possibility is that in anti-cyclones all long-term trends relate to phase transitions consistent with the mixed-phase cloud feedback. However, this seems unlikely given the extensive analysis performed by Terai et al. (2016) demonstrating a substantial contribution to cloud optical depth variability

in the Southern Ocean from variability in estimated inversion strength (EIS, Wood and Bretherton (2006)), and a lesser contribution linearly related to SST. It is also worth noting that SST was found to decrease or increase cloud optical depth, depending on the observational data and season examined(Terai et al., 2016). Thus, it is unlikely that all the trend in anti-cyclonic regions is related to phase transitions and it is more likely that the trend in these regions is dominated by trends in boundary-layer cloudiness consistent with enhancing inversion strength(Terai et al., 2016), which is a well-quantified feature

of boundary-layer cloud cover(Wood and Bretherton, 2006;Klein and Hartmann, 1993). We reserve a more complete examination of cloud variability composited around both high and low pressure centers for a future paper and will focus on examining low pressure centers in the present work.

As discussed above, it is likely that in a warming world the change in cyclone vorticity, and thus wind speed will be relatively slight. If we assume that the distribution of wind speed remains unchanged, that frequency of occurrence of cyclones

remains unchanged, and that WVP increases by 6%/K (Fig. 6c) we can estimate the change in WCB moisture flux in the NH (0.22 mm/day, Fig. S7) and SH (0.20 mm/day Fig. S8) consistent with a uniform 1K warming. Assuming that the WCB-LWP$_{CM}$ relationship remains unchanged in a warmed climate this new distribution of WCB moisture flux would yield an increase of 3.42 g/m$^2$ in LWP in the NH and a 3.71 g/m$^2$ increase in the SH (Fig. S7 and Fig. S8). We offer this estimate in order to provide an approximate scale to the potential of the WVP-mediated changes in extratropical cyclone LWP in a

warming climate. An estimate of the change in reflectivity consistent with this change in cyclone LWP will be offered below.

Comparison of the trend in Southern Ocean LWP$_{RM}$ for the models listed in Table 2 are shown in Fig. 7b. Despite the models being run in AMIP-mode with prescribed SSTs, there is significant variability in the trend in LWP$_{RM}$ in the period 1992-2015 (note that CFMIP2 models simulated 1979-2008). The GCMs all show a positive trend that is significant at 95% confidence, but generally under-predict the strength of the trend. It is also interesting to note that the trend in LWP$_{RM}$ across



the Southern Ocean is almost completely explained by WCB moisture flux variability in all the GCMs. In the following section we will revisit this puzzle and use spatial variability within cyclone composites and cloud top phase to attempt to disentangle the contributions of ice-to-liquid transitions and WCB moisture flux.

As mentioned above, this analysis relies on the assumption that the relationship between WCB moisture flux and
LWP is invariant under warming. At the time of writing the only GCMs considered in this study that have simulated a global increase in temperature (outside of the observational record) are the GCMs participating in CFMIP2 (see Table 2). The CFMIP2 GCMs performed a set of simulations where the specified SST in the atmosphere-only (AMIP) runs was increased by 4K (AMIP+4K). The CFMIP2 GCMs represent a wide array of different relationships between WCB moisture flux and cyclone LWP and it is hoped that even though we have limited our analysis of cyclone behavior in a warmed climate to these
models, it still provides insight into the broader collection of models examined in the rest of this study.

Comparison of the WCB moisture flux-LWP relationship (see Fig. 3) between the AMIP and AMIP+4K CFMIP2 simulations shows that they are fairly similar in the SH (Fig. 8). Only IPSL-CM5A-LR has a substantially different relationship between $LWP_{CM}$ and WCB moisture flux in the AMIP and AMIP+4K simulations. Examination of the NH shows that all the models display a downward shift in the WCB moisture flux-LWP relationship in the warmed simulations (Fig. S9). It is unclear
why the WCB moisture flux-LWP relationship in the NH shifts downward, while it shifts upward in the SH in only one of the GCMs. At least in the SH this upward shift is conceptually consistent with a decrease in precipitation efficiency due to decreased ice-phase precipitation. In this case an increase in cyclone LWP would be in line with the necessity of balancing precipitation out of the cyclone and moisture flux into the cyclone(McCoy et al., 2018b).

Changes in extratropical cyclones in a warming climate may affect the relationship between WCB moisture flux and
LWP. However, moisture flux changes still explain most of the difference in cyclone LWP between the AMIP and AMIP+4K simulations examined here. For each GCM a linear regression model of the form $LWP_{CM} = a \cdot WCB + b$ is fit using the variability in the present-day AMIP simulations. Regression models are trained independently in each hemisphere. Cyclone LWP in the AMIP+4K simulations is predicted based on the regression model and the WCB moisture flux in the AMIP+4K simulations. That is to say, if we know the relationship between moisture flux and cyclone LWP in the current climate, and we
know how moisture flux changes, then can we predict the change in cyclone LWP?

Changes in WCB moisture flux explain the majority of cyclone LWP difference between the AMIP+4K and AMIP simulations (Fig. S10). Over 80% of the difference in southern hemisphere cyclone LWP between AMIP and AMIP+4K is explained by differences in WCB moisture flux in four out of five of the models. The change in cyclone LWP in IPSL-CM5A-LR is 30% greater than the change predicted by WCB moisture flux alone, consistent with a phase-transition-driven increase
in LWP. Differences in northern hemisphere cyclone LWP between AMIP and AMIP+4K are within 25% of the prediction based on the AMIP WCB moisture flux-LWP relationship and the difference in WCB moisture flux (Fig. S11).

Changes in WCB moisture flux dominate the change in Southern Ocean cyclone LWP in the CFMIP2 models between AMIP and AMIP+4K. We also examine changes in the components of WCB moisture flux: WVP and wind speed. The change between AMIP and AMIP+4K agrees with the sensitivity inferred from the interannual variability (Fig. 6bc). As cyclones shift





poleward their mean wind speed increases, and as SSTs rise WVP increases (Fig. S12). As prescribed by the AMIP+4K simulations the SST rises in both hemispheres. The response in mean cyclone position is varied and difficult to interpret in the context of a green house gas-induced warming due to the fixed SST imposed in these simulations. However, it does appear that the relationship between inter-annual anomalies in average extratropical cyclone latitude and wind speed from the current
climate holds in a warmed climate.

In summary, we find that most of the cyclone LWP trend in the SH observational record can be explained by a steady increase in WCB moisture flux, as opposed to a transition to less-glaciated clouds. We support this result by contrasting CFMIP2 AMIP and AMIP+4K simulations. More than 70% of the difference in cyclone LWP between these simulations can be explained by changes in WCB moisture flux. In the next section we will utilize observations of cloud-top phase to further
examine how cloud glaciation might affect cyclone LWP.

### 3.4 Changes in LWP and cloud-top phase

In the preceding sections we have investigated the link between large-scale meteorology, as characterized by WCB
moisture flux, and extratropical cyclone LWP averaged to a cyclone-mean, or regional scale. The moisture flux along the WCB explains a great deal of the variability in cyclone LWP in models and observations and that the decadal trend in SH $LWP_{RM}$ seems to be largely related to WCB moisture flux. Here we will use observations of cloud-top phase to examine whether any variability in extratropical cyclone LWP can be linked to a transition from ice to liquid consistent with a warming signal (that is to say, phase transitions consistent with the mixed-phase cloud feedback).
As noted above, WVP depends on SST via Clausius-Clapeyron, making it difficult to empirically disentangle SST-driven ice-to-liquid transitions from moisture flux driven variability. Here we will examine the response of LWP within the cyclone composite ($LWP_{ij}$) based on multiple linear regression on WCB moisture flux into the cyclone and $SST_{ij}$. The regression model considered here is

$$LWP_{ij} = a_{ij} \cdot WCB + b_{ij} \cdot SST_{ij} + c_{ij}$$                                    [6]

where the subscripts i and j refer to areal averages within the composites in the longitudinal and poleward directions (see Table 1). Each averaging region is approximately 200 kmx200 km. Values for the coefficients are calculated by fitting the regression model across cyclones in each averaging region $ij$. The coefficient values of $a_{ij}$ and $b_{ij}$ are shown in Fig. 9. Unsurprisingly, there is a strong positive relationship between WCB moisture flux and $LWP_{ij}$ throughout the cyclone (e.g. the coefficient $a_{ij}$ in Eq. 6). Increasing $SST_{ij}$ tends to covary with increased $LWP_{ij}$ in the part of the composite poleward of the low and with
decreased $LWP_{ij}$ in the portion of the composite equatorward of the low (e.g. $b_{ij}$).

Most of the variability in the sensitivity of $LWP_{ij}$ to $SST_{ij}$ is in the latitudinal direction, while sensitivity to WCB peaks near the origin. To simplify our presentation and compare across GCMs and observations, we show the regression coefficients for the GCMs in Table 2 and the observations averaged in the longitudinal direction (Fig. 10). Comparison between



models and observations show that there is variability between models and observations regarding the sensitivity of $LWP_{ij}$ to WCB moisture flux into the cyclone (Fig. 10ac), which is consistent with the range of slopes shown in Fig. 3a. However, the relation between $LWP_{ij}$ and $SST_{ij}$ within the composite is fairly similar across models (Fig. 10bd). Increasing SST in the equatorward part of the composite tends to covary with decreasing $LWP_{ij}$ and increasing $SST_{ij}$ in the poleward part of the

composite covaries with increases in $LWP_{ij}$.

This negative relationship between local changes in $SST_{ij}$ and $LWP_{ij}$ within the part of the composite that is equatorward of the low agrees with previous studies showing break up in midlatitude stratocumulus with advection over warmer SSTs due to decoupling of the subcloud layer (Norris and Iacobellis, 2005), and is consistent with the prevailing hypothesis regarding warm clouds in the sub-tropical trade cumulus and stratocumulus regions (Klein et al., 2017). It is possible

that the poleward enhancement in $LWP_{ij}$ in response to enhancement in $SST_{ij}$ may relate to shifts from ice to liquid cloud, but it might also relate to other meteorological controls on cloud cover and thickness(Grise and Medeiros, 2016).

As shown in Bodas-Salcedo (2018) and Bodas-Salcedo et al. (2016), the effect of changes in LWP are highly dependent on the cloud regime that they are occurring in. In particular, overlying cloud can act to blunt the effect of changes in LWP on top of atmosphere radiation. Do the changes in LWP inferred from Fig. 9 translate into a meaningful change in

reflected shortwave? We offer an approximate calculation of the change in reflected shortwave consistent with Fig. 9cd using observations from CERES to account for the effects of masking by overlying ice cloud. Daily-mean all-sky albedo from CERES SYN1DEG(Doelling et al., 2016;Wielicki et al., 1996) was created from 3-hourly data from 2003-2015 where the solar zenith angle does not exceed 45° (see McCoy et al. (2018b) for a full discussion of this data). Regression of albedo on LWP variability gives an empirical relationship between LWP and albedo (Fig. S13). Radiative fluxes are more readily

comparable to previous studies of cloud feedbacks. Thus, the change in albedo is scaled by the annual mean insolation taken from the CERES EBAF-TOA edition 4 dataset (Loeb et al., 2009) averaged over 30-80° to give the change in $Wm^{-2}$ per change in LWP. While empirical, this is a relatively simple way to examine the effects of cloud masking.

LWP is always positively correlated with albedo (Fig. S13). At zeroth order we expect this based on the robust positive relationship between cloud fraction and all-sky albedo(Bender et al., 2011a;Bender et al., 2017), and remembering that

microwave LWP is the average of in-cloud liquid and clear sky. If we multiply the relationships for the SH shown in Fig. 9cd by the slope of the regression between LWP and albedo, then this gives the change in albedo across the cyclone composite consistent with Eq. 6 and a unit increase in WCB moisture flux or SST. WCB moisture flux increases by approximately 0.2 mm/day for a 1K increase in SST in the SH if wind speed is held constant and WVP increases following Clausius-Clapeyron (Fig. S8). Thus, we scale the change in albedo per unit change in WCB moisture flux by 0.2 mm/day to give a change in albedo

related to changes in WCB moisture flux consistent with a 1K SST increase (assuming no change in wind speed). In the context of Eq. 6 the net change in reflected shortwave that is implied by a 0.2 mm/day increase in WCB moisture flux is 0.87 $Wm^{-2}$ and reflected shortwave decreases by 0.23 $Wm^{-2}$ for a 1K SST increase (Fig. S14). Again, these empirical calculations are simplistic and are only intended to approximate the effect of cloud masking, which has been identified as a key factor in cloud feedbacks in midlatitude cyclones(Bodas-Salcedo, 2018). A more precise estimate of changes in cyclone albedo accounting



for cloud masking is reserved for future work. Overall, we find that the changes in LWP that are empirically linked to changes in WCB moisture flux and SST in the multiple regression shown in Eq. 6 translate to reasonably large negative and positive feedbacks, respectively, providing that the current relationship between LWP and albedo within cyclones holds in a warming world. These implied feedbacks may be contrasted with the zonal-mean cloud feedbacks from the CFMIP2 and CFMIP1

models, with a strongest value for the dipole of -2 Wm$^{-2}$ (Zelinka et al., 2013;Zelinka et al., 2016).

We have shown that LWP changes associated with changes in SST and moisture flux have the capability to change reflected shortwave. Is this increase in $LWP_{ij}$ with increasing $SST_{ij}$ in the poleward half of the composite from phase transitions? We examine the sensitivity of cloud top phase to $SST_{ij}$ and WCB to see if there is any consistency in regions where clouds become more liquid dominated, and regions where the $LWP_{ij}$ sensitivity to $SST_{ij}$ suggests a phase transition. Cloud top

phase is measured by the AIRS instrument during the period 2003-2015. As discussed in the methods section, when the cloud is broken, mixed-phased, or possibly supercooled liquid the infrared signature becomes weak and the cloud top is flagged as unknown by AIRS. Here, we examine the probability that a given cloud-top phase (liquid, ice, or unknown) was detected by AIRS given that any phase detection was made in a cyclone composite framework (Fig. 11). We perform the same analysis on the probability of a cloud top being flagged as liquid, ice, or unknown as performed on LWP. We examine how phase depends

on WCB moisture flux, and how it depends on $SST_{ij}$. This is done analogously to the analysis performed above in the context of multiple linear regression

$$p(x)_{ij} = a_{ij} \cdot WCB + b_{ij} \cdot SST_{ij} + c_{ij} \qquad [6]$$

where $p(x)_{ij}$ is the probability of a cloud top phase being an arbitrary phase $x$ (ice, liquid, or unknown) given that a phase detection was made. The coefficients from performing this regression across cyclones in the SH and NH are shown in Fig. 12.

Overall, the effect of increasing moisture flux into the cyclone is to increase the frontal cloud which is ice-topped (Fig. 12c and Fig. 13c). The ice cloud is anti-correlated with SST- as one would intuitively expect (Fig. 12d and Fig. 13d). However, most of this is along the comma-shaped frontal region, consistent with Bodas-Salcedo (2018).

While the SST-dependence of $LWP_{ij}$ is quite similar across models and observations in both hemispheres (Fig. 10bd), the dependence of observed cloud-top phase on $SST_{ij}$ is very different in the NH and SH. In the NH, the probability of liquid-

topped clouds increases with increasing SST in the poleward part of the composite (Fig. 12b). Toward the equator, increasing SST increases the fraction of unknown cloud tops (Fig. 12f). In the SH increasing SSTs covaries with enhancement in the prevalence of unknown cloud tops at the expense of liquid and ice fraction across the entire cyclone composite (Fig. 13bdf).

Increasing liquid fraction in the poleward half of NH cyclones over warmer SSTs (Fig. 12b) is consistent with transitions from a more ice-dominated to a more liquid-dominated state. The covariance of $LWP_{ij}$ and $SST_{ij}$ in these regions

(Fig. 9b) appears to bear out this explanation. This may simply reflect an increase in liquid at the expense of ice, or it may reflect an increase in overall condensate via suppression of the efficient depletion of cloud condensate via ice-phase precipitation(Field and Heymsfield, 2015).

The decrease in both liquid and ice fraction and the increase in unknown cloud tops over warmer SSTs in equatorward portion of NH cyclones and in the entirety of SH cyclone composites is perplexing. It may be linked to transitions from closed



to open mesoscale cellular convection(McCoy et al., 2017c;Norris and Iacobellis, 2005) leading to weaker IR signals and thus unknown cloud top-phase classification from AIRS (Nasiri and Kahn, 2008;Kahn et al., 2011). It is unclear why this behavior only takes place in the equatorward parts of NH cyclones, while it occurs across the entire cyclone composite in the SH. One possibility is that the Southern Oceans are much more dominated by supercooled liquid cloud(Chubb et al., 2013;Kanitz et al.,

2011;Hu et al., 2010;Tan et al., 2014) and they do not have the same phase transition sensitivity as the NH oceans. This seems consistent with the lower sensitivity of poleward LWP within SH cyclones to local $SST_{ij}$ (Fig. 10bd). Overall, the magnitude of the decrease in liquid-topped clouds with increased $SST_{ij}$ is relatively slight in SH. The magnitude of the increase in unknown tops with increasing $SST_{ij}$ in the SH is relatively similar to magnitude of the increase in liquid-topped clouds in the NH. Finally, the decrease in ice-topped cloud across the cyclones in both NH and SH is fairly similar with an ~0.075%/K

decrease implied by covariability in both (Fig. S15b).   A higher proportion of supercooled liquid phase clouds in the SH is also consistent with an increasingly ambiguous spectral signature in the mid-infrared window region as the index of refraction for supercooled liquid is known to be temperature-dependent (Rowe et al., 2013) and is not accounted for in the current AIRS thermodynamic phase algorithm.

        To summarize, we investigate the possibility that an ice to liquid transition consistent with the mixed-phase feedback

proposed in other studies may account for LWP trends unrelated to trends in WVP, which are ultimately driven by Clausius-Clapeyron and increasing SST. Disentangling contributions from moisture flux and phase transitions is a difficult problem because WVP, SST, and WCB moisture flux covary. We leverage the cloud-top phase retrievals from AIRS and the geographic distribution of variability within the cyclone composite to try and detach variability associated with moisture flux from variability associated with phase transitions. A great deal of the inter-model variability appears to be owing to variability in

the portrayal of the WCB moisture flux-LWP relationship, and it is found that models and observations are in surprisingly good agreement regarding the covariability of $LWP_{ij}$ to $SST_{ij}$ in the context of a multiple linear regression of $LWP_{ij}$ on WCB and $SST_{ij}$(Fig. 10). There is an increase in $LWP_{ij}$ with increasing $SST_{ij}$ in the poleward part of the cyclone that is spatially consistent with the region where $SST_{ij}$ and liquid-topped cloud fraction are positively correlated in the NH. This is supportive of this sensitivity of $LWP_{ij}$ to $SST_{ij}$ being linked to phase transitions. Ultimately, the cancellation of a positive covariability

between $SST_{ij}$ and $LWP_{ij}$ in the poleward region of the cyclone and a negative covariability in the equatorward regions leads to an overall slight negative covariability between $SST_{CM}$ and $LWP_{CM}$ is considered (Fig. 7).

        Unfortunately, many of the cloud tops in the Southern Ocean cannot be confidently classified as either liquid or ice by AIRS (Thompson et al., 2018). This makes it difficult to establish whether cyclones are transitioning to a more liquid-dominated state.   In the SH, increasing SST covaries with increasing unknown phase identifications. One possibility is that

increasing SST alters the mesoscale cellular convection leading to open cells(McCoy et al., 2017c;Norris and Iacobellis, 2005), which AIRS cannot identify. Another possibility is that the SH ocean is dominated by supercooled liquid cloud, and thus does not become any more liquid with warming(Hu et al., 2010). This corresponds to decreasing LWP with increasing SST in both observations and models. This may point toward the pathway: higher SST, more cumuliform cloud, and more broken cloud(Norris and Iacobellis, 2005), which is also consistent with other empirical studies of low cloud cover in the



subtropics(Klein et al., 2017). Further clarity in the cloud phase information from AIRS may require additional algorithm development to exploit hyperspectral infrared radiances to classify supercooled liquid and mixed phase spectral signatures (e.g., Kahn et al. (2011); Rowe et al. (2013)), including the impacts of sub-pixel horizontal variations of liquid and ice phase mixtures (e.g., Thompson et al. (2018)).

## 4 Conclusions

We have examined the behavior of extratropical cyclones (centered 30°-80°) in models and long-term microwave observations for the period 1992-2015. The central tool used in this study is the ability to characterize cyclones by their meteorology using the warm conveyor belt (WCB) moisture flux. As the moisture flux along the WCB increases the liquid water path (LWP)

within the cyclone increases. We find that the WCB moisture flux explains 28-42% of anomalous monthly, regional variability in LWP across different ocean basins (Fig. 5a).

This relationship between the synoptic state, as characterized by WCB moisture flux, and the cyclone LWP appears across observations and models. The ability of GCMs to reproduce this relationship is examined using an array of models from high-resolution simulations within the convective grey zone to coarse resolutions typical of fully-coupled GCMs performing

climate integrations. It is found that all the models considered here reproduce the dependence of LWP on moisture flux, but their sensitivities vary by a factor of two around the observed sensitivity. There did not appear to be a strong, systematic dependence of this relationship on resolution (Fig. 3). Overall, we suggest that this relationship between moisture flux and LWP within cyclones should be used in the evaluation of the physicality of GCMs. Further, we find that simulations that do not include a convective parameterization (NICAM, ICON, and UM-CASIM) tend to have much more similar relationships

between WCB moisture flux and cyclone LWP, indicating that parametric uncertainty within convective parameterizations may contribute to uncertainty in this relationship across GCMs.

Variability in WCB moisture flux appears to drive variability in midlatitude cyclone LWP. What in turn drives variability in WCB moisture flux? The WCB moisture flux is the product of water vapor path (WVP) and wind speed at 10 meters ($WS_{10m}$) averaged within the cyclone. WVP is strongly coupled to SST via Clausius-Clapeyron, and $WS_{10m}$ enhances

in cyclones as they move poleward through their life cycle (Fig. 5bc). This feature exists in both observations and models (Fig. 6bc). These relationships also hold in the CFMIP2 models when SSTs are increased by 4K (AMIP+4K) (Fig. S12). We propose two extratropical cloud feedbacks within cyclone systems: a Clausius-Clapeyron mediated local feedback, and a dynamical feedback related to shifts in the storm track and changes in cyclone development in response to warming. The former feedback is in line with the feedback proposed in Betts and Harshvardhan (1987), but expressed in the framework of a midlatitude

cyclone, which imposes a structure on the derivative of the moist adiabat with respect to temperature. The sign of the wind speed-driven feedback appears to be uncertain in a warming climate. Between the CFMIP2 AMIP and AMIP+4K simulations the midlatitude cyclones shifted poleward and equatorward in different models (Fig. S12), but the relationship between latitude



and wind speed between AMIP and AMIP+4K mirrored the relationship inferred by variability within AMIP. This relationship creates a pathway between synoptic-scale dynamics and the cloud feedback. A simple calculation holding wind speed and cyclone frequency of occurrence fixed and assuming that WVP changes by 6% estimates an increase in cyclone LWP of 3.42 g/m$^2$ in the NH and 3.71 g/m$^2$ in the SH (Fig. S7 and Fig. S8). An empirical calculation of brightening estimates that this would

equate to a brightening of 0.87 Wm$^{-2}$ within Southern Ocean cyclones (Fig. S14), which is an appreciable fraction of the overall shortwave cloud feedback(McCoy et al., 2016;Zelinka et al., 2013).

We find that there is support for a negative feedback described above by examining decadal trends in LWP within Southern Ocean cyclones (Fig. 7). Most of the observed trend over the period 1992-2015 can be explained by increased WCB moisture flux, which is primarily driven by increasing WVP. Further, we find that analysis of simulations performed with

observed SSTs enhanced by 4K show that differences in WCB moisture flux can explain the majority of the change in simulated cyclone LWP between the current climate and the warmed climate (Fig. S10,11). This supports the idea that our understanding of the relationship between WCB moisture flux and cyclone LWP in the current climate may be used to understand cloud feedbacks.

The majority of observed and modeled trends in SH cyclone LWP in the current climate can be explained by changes

in WCB moisture flux. Similarly, the majority of the simulated change in cyclone LWP between the current climate and a warmed climate can be explained by changes in WCB moisture flux. Do changes in cloud phase play a role in altering LWP within extratropical cyclones? While changes in WCB moisture flux explain a great deal of LWP variance within cyclones, there is a residual signal in LWP related to SST leading to increased LWP in the poleward half of the composite (Fig. 10bd). In the half of the composite that is equatorward of the low the LWP decreases with increasing SST, in keeping with previous

studies(Norris and Iacobellis, 2005). Utilizing cloud top phase data from AIRS we show that changes from ice to liquid cloud could contribute to increasing LWP with increasing SST in the poleward half of the composite (Fig. 12b and Fig. 13b). In the equatorward part of the composite unknown phase (broken or mixed-phase) cloud tops become more frequent as SST increases across the NH (Fig. 12f). In the SH unknown phase cloud tops increase at the expense of liquid and ice in all parts of the composite (Fig. 13f, and Fig. S15b). This may be consistent with breakup of stratocumulus over warmer SSTs.

**Acknowledgements**

CK acknowledges the support of Japan Society for the Promotion of Science (JSPS) KAKENHI Grant Number 17H04856 and the support of MEXT through Integrated Research Program for Advancing Climate Models (TOUGOU program), Stragetic Programs for Innovative Research (SPIRE) Field 3 using the K computer resources (Proposal number hp120279, hp130010,

and hp140219), and the FLAGSHIP2020 within the priority study4 (Advancement of meteorological and global environmental predictions utilizing observational "Big Data"). A portion of this work was carried out at the Jet Propulsion Laboratory (JPL), California Institute of Technology, under a contract with the National Aeronautics and Space Administration. The effort of



MDZ was funded by the RGCM Program of the U.S. Department of Energy (DOE) and was performed under the auspices of the U.S. DOE by Lawrence Livermore National Laboratory under contract DE-AC52-07NA27344. GE acknowledges support from the NASA MEaSUREs program (via subcontract with the Jet Propulsion Laboratory; Grant no. GG008658). DTM and PRF acknowledge support from the PRIMAVERA project, funded by the European Union's Horizon 2020 programme, Grant

Agreement no. 641727. We acknowledge use of the MONSooN system, a collaborative facility supplied under the Joint Weather and Climate Research Programme, a strategic partnership between the Met Office and the Natural Environment Research Council.

**Data availability**

The AIRS version 6 data sets were processed by and obtained from the Goddard Earth Services Data and Information Services Center (http://daac.gsfc.nasa.gov/). The AIRS data used in this study is cited as: AIRS Science Team/Joao Texeira (2013), AIRS/Aqua L2 Support Retrieval (AIRS+AMSU) V006, Greenbelt, MD, USA, Goddard Earth Sciences Data and Information Services Center (GES DISC), Accessed July 2017, doi:10.5067/Aqua/AIRS/DATA207. MERRA2 data was downloaded from the Giovanni data server. CERES data was downloaded through the ceres.larc.nasa.gov ordering
interface.

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

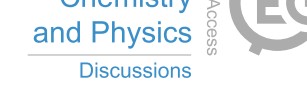

**Table 1 Acronyms and subscripts used in this work.**

| Acronym | Defintion |
|---|---|
| CCN | Cloud condensation nuclei |
| CDNC | Cloud droplet number concentration |
| CM | Cyclone mean within a 2000 km radius of the low pressure center |
| GCM | Global climate model |
| LWP | Liquid water path |
| ij | Mean within each averaging region of the cyclone |
| NH | Northern hemisphere |
| RM | Regional mean of individual cyclone means |
| SH | Southern hemisphere |
| SLP | Sea level pressure |
| SO | Southern Ocean |
| SST | Sea surface temperature |
| WCB | Warm conveyor belt |
| WVP | Water vapor path |
| WS$_{10m}$ | Wind speed at 10 meters |



**Table 2 Brief descriptions of the models used in this study. The label in the left column is used in some figures for brevity in labelling. The observations used in this study are discussed more completely in the methods section.**

| Label | Name | Approximate Atmospheric Resolution | References | Time Period |
|---|---|---|---|---|
| A | Observations | ~ | See methods | 1992-2015 |
| B | HadGEM2-A[1] | 1.25°x1.875°~ 139kmx208 km | (Collins et al., 2011;Martin et al., 2011) | 1979-2008 |
| C | IPSL-CM5A-LR[1] | 1.8947x3.75° ~211kmx417km | (Dufresne et al., 2013) | 1979-2008 |
| D | MIROC5[1] | 1.4008°x1.40625° ~156kmx156km | (Watanabe et al., 2010) | 1979-2008 |
| E | IPSL-CM5B-LR[1] | 1.8947°x3.75° ~211kmx417km | (Hourdin et al., 2013) | 1979-2008 |
| F | CNRM-CM5[1] | 1.4008°x1.40625° 156kmx156km | (Voldoire et al., 2013) | 1979-2008 |
| G | NICAM | 14km | (Kodama et al., 2015) | 1979-2007 |
| H | EC-Earth3[2] | 60km | (Haarsma, 2018) | 1989-2014 |
| I | EC-Earth3-HR[2] | 25km | | 1989-2014 |
| J | CNRM-CM6-1[2] | 150km | (Roehrig, 2018) | 1989-2014 |
| K | CNRM-CM6-1-HR[2] | 50km | | 1989-2014 |
| L | HadGEM3-GC31-LM[2] | 130km | (Williams et al., 2018) | 1989-2014 |
| M | HadGEM3-GC31-MM[2] | 60km | | 1989-2014 |
| N | HadGEM3-GC31-HM[2] | 25km | | 1989-2014 |
| ~ | ICON | 10km | (Giorgetta et al., 2018) | |



| ~ | UM-CASIM | 0.088°x0.059° 10kmx7km | (McCoy et al., 2018b;Hill et al., 2015) | |
|---|---|---|---|---|
| | [1]CFMIP2 [2]PRIMAVERA | | | |

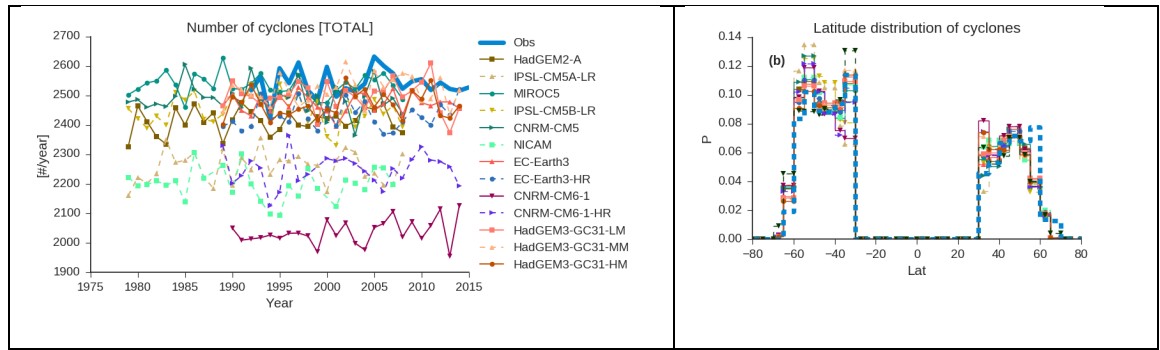

5    **Fig. 1 The distribution of daily-mean cyclone centers analyzed in this study. (a) The number of extratropical (30°-80°) cyclone centers identified each year in daily-mean data in the MERRA2 reanalysis and different global models and (b) the latitudinal distribution of cyclone centers. Only cyclone centers over water and where more than 50% of the cyclone center is over ice-free ocean are considered. In (b) the number of cyclone centers identified in each 5° latitude bin is divided by the total number of cyclones. Each cyclone center identified in the daily-mean data is considered independently. SH and NH cyclones are shown combined. In (a) the**
10    **number of cyclones in a year is rescaled assuming a 365-day year because some of the GCMs have a 360-day year.**





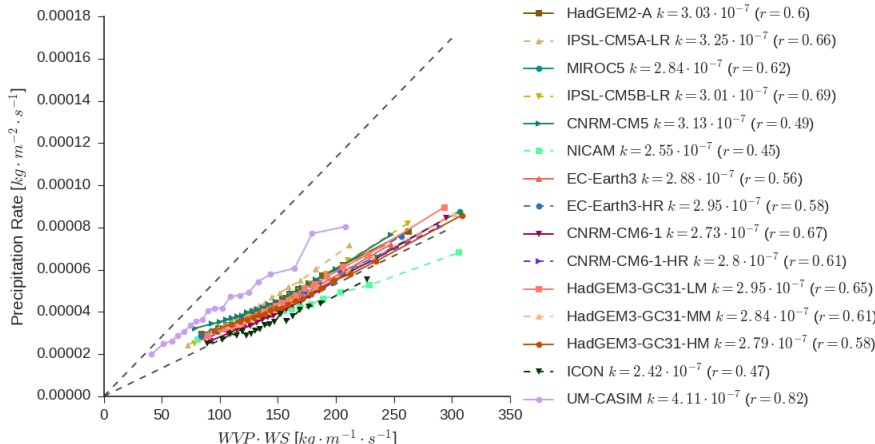

**Fig. 2 Cyclone-mean precipitation rate versus WCB moisture flux for the global models examined in this study. The *k* parameter (Field et al., 2011) for each model is noted in the legend along with the correlation between WCB moisture flux and cyclone-mean precipitation rate. The *k* parameter is calculated as the slope of the relationship between the product of WVP and $WS_{10m}$ and the**
5 **precipitation rate. Dashed lines show the observational bounds on *k* (Field et al., 2011;Naud et al., 2018). For ease of visualization the precipitation rates for each GCM are shown averaged into 19 quantiles of $WVP_{CM}*WS_{10mCM}$.**

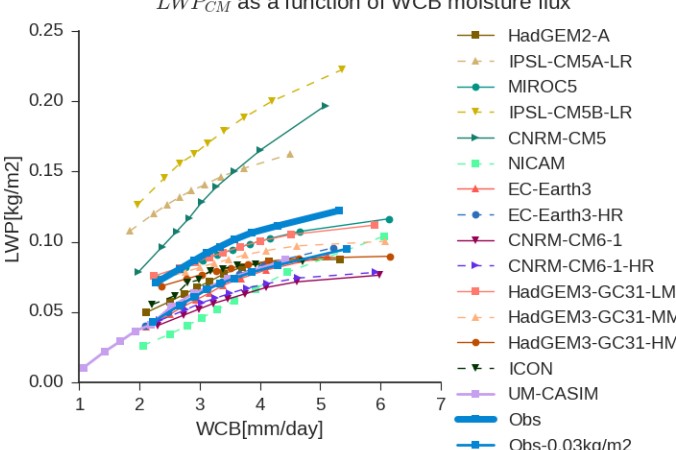

**Fig. 3 Cyclone-mean LWP ($LWP_{CM}$) as a function of WCB moisture flux in models and observations. $LWP_{CM}$ is shown averaged**
10 **into 9 equal quantiles for the observations and each GCM. The maximum bias in the observations (~0.03kg/m2) is shown as a lighter blue line.**



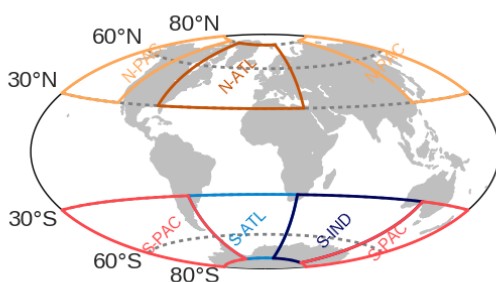

**Fig. 4 Averaging regions considered in Fig. 5. Labels refer to the North Pacific (N-PAC), North Atlantic (N-ATL), South Pacific (S-PAC), South Atlantic (S-ATL), and South Indian (S-IND) oceans.**





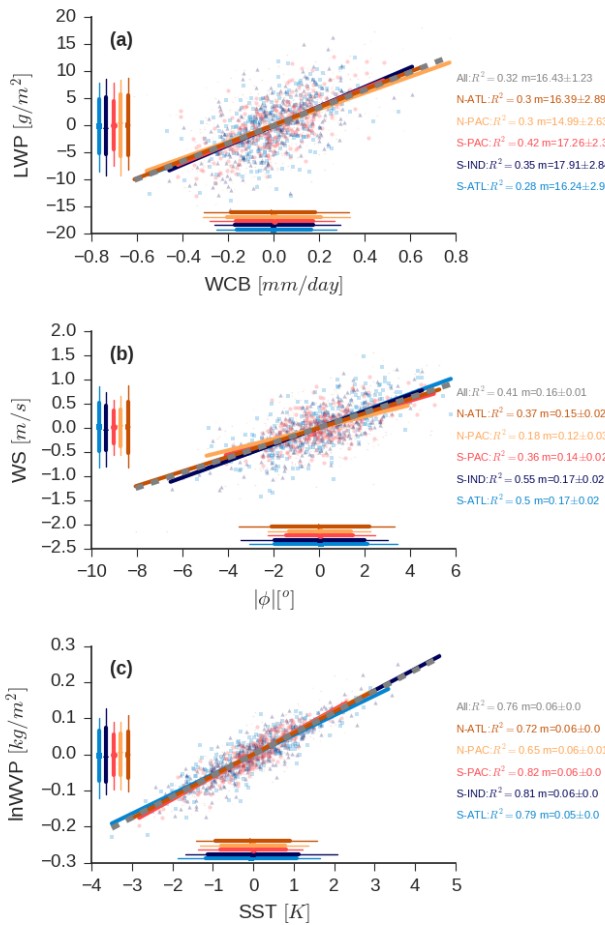

**Fig. 5 Observed monthly- and regional-mean anomalies in extratropical cyclone properties in the SH and NH oceans. (a) Cyclone monthly-mean LWP$_{RM}$ as a function of WCB$_{RM}$; (b) regional- and monthly-mean wind speed as a function of regional- and monthly-mean cyclone absolute (poleward) latitude; (c) monthly-mean ln(WVP)$_{RM}$ as a function of SST$_{RM}$. Each data point in the plot represents the monthly- and regional-mean anomaly in a given variable within extratropical cyclones relative to climatology. The R$^2$ and best fit line are listed for each subplot and for each ocean region. The R$^2$ of all monthly- and regional-mean anomalies is also noted (the variability in regional- and monthly-mean anomalies are weighted equally between regions to calculate the overall R$^2$). Bars on the sides of the plot show the mean (marker), standard deviation (thick lines) and 90$^{th}$ percentile range (thin lines) of monthly- and regional-mean anomalies for each region.**





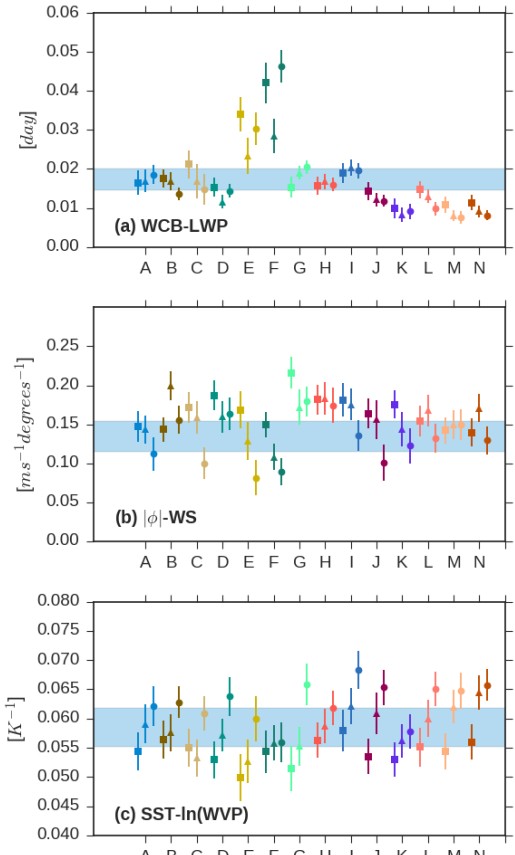

**Fig. 6 The slope of the best fit line between monthly- and regional-mean anomalies of different cyclone-mean properties. Symbols denote different SH ocean basins (South Atlantic:squares, South Indian:triangles, and South Pacific:circles). Model colors are as in Fig. 1. Each model is labelled with a letter on the ordinate (see Table 1). The observations are shown as 'A'. The 95% confidence on the slope is noted for each basin. The shaded area shows the 95% confidence on the mean of the observed slope based on the SH ocean basins. (a) shows the slope of the regression of cyclone LWP$_{RM}$ on WCB moisture flux, (b) shows the regression slope of the mean wind speed in cyclones on mean poleward latitude; and (c) shows the regression slope of ln(WVP)$_{RM}$ on SST$_{RM}$.**



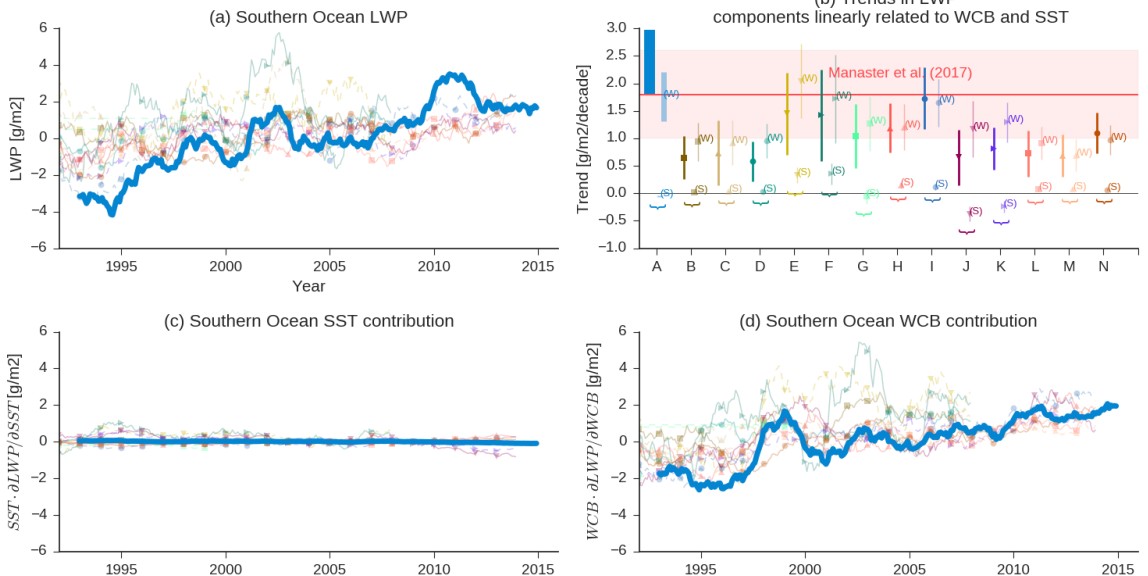

**Fig. 7 (a) SH cyclone LWP_RM in observations (thick blue line) and models (colors as in Fig. 1) where cyclones are centered between 44.5°S and 59.5°S. A 2-year running mean has been applied to simplify the plot. (b) The red shaded area shows the zonal-mean Southern Ocean LWP trend calculated in Manaster et al. (2017) (Southern Ocean defined as 44.5°S-59.5°S therein). Trends in cyclone LWP_RM from observations and models are shown in dark colors (where LWP_RM is calculated using cyclones centered in the same region as Manaster et al. (2017)). Models and observations are labelled by a letter on the ordinate (see Table 2). Observations are labelled as 'A'. The 95% confidence in each trend is shown using errorbars. A multiple linear regression of LWP_CM on SST_CM and WCB moisture flux is used to partition the trend into contributions from WCB and SST changes. The trend in LWP_RM predicted by the regression model (Eq. 4) and changes in SST_RM is shown in (c), and the trend in LWP_RM associated with WCB is shown in (d). The trend from models and observations consistent with their multiple linear regression models and changes in SST_RM and WCB is shown in (b) using lighter colors and labelled as (S) for SST, and (W) for WCB.**




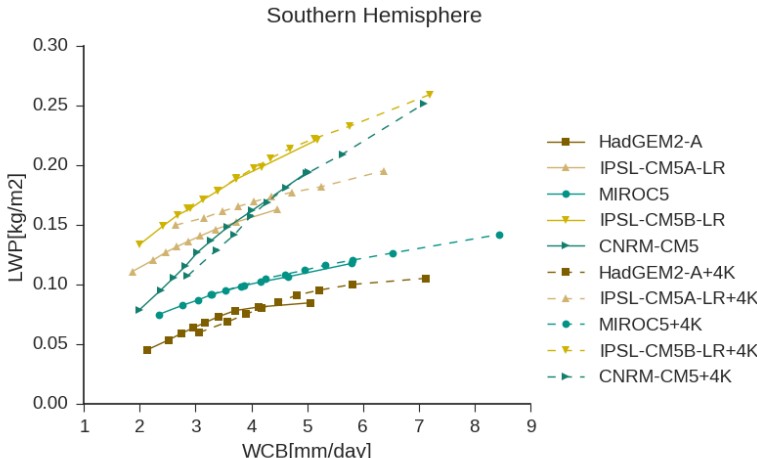

**Fig. 8 As in Fig. 3, but contrasting the AMIP and AMIP+4K simulations in the CFMIP2 simulations.**

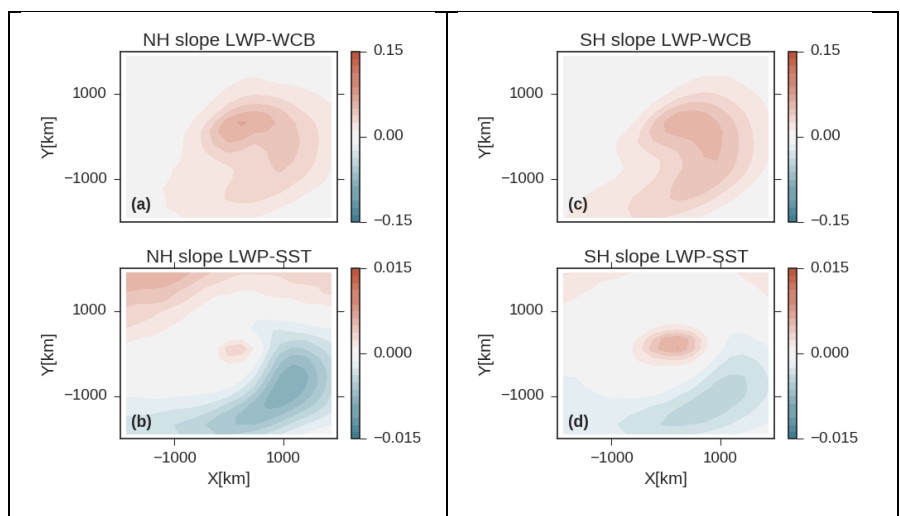

**Fig. 9 The multiple linear regression coefficients (Eq. 6) relating observations of LWP$_{ij}$ to SST$_{ij}$ and WCB moisture flux in the NH (a and b) and SH (c and d). Multiple linear regression is used to partition LWP$_{ij}$ into contributions from SST$_{ij}$ and WCB moisture flux. (a) and (c) show the slope of the linear regression between the WCB moisture flux into the cyclone and LWP$_{ij}$ within the composite (units are kg mm day$^{-1}$m$^{-2}$). (b) and (d) show the regression coefficient relating SST$_{ij}$ and LWP$_{ij}$ (kg m$^{-2}$K$^{-1}$). Note that**
10   **SH cyclones have been flipped vertically so that the top of the plot is to the pole to facilitate comparison to NH cyclones.**



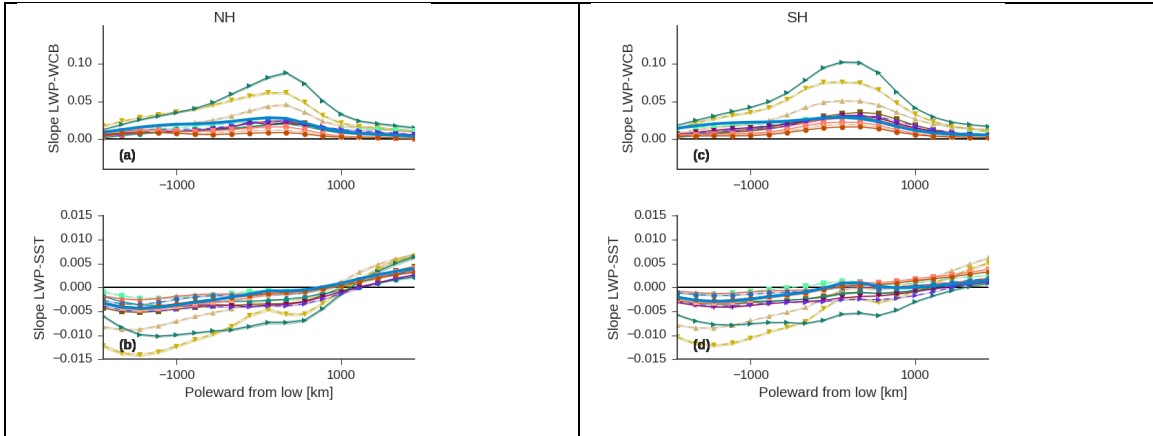

**Fig. 10** As in Fig. 9, but showing the multiple linear regression slopes averaged zonally (Eq. 6) across the composite. The x-axis shows distance from the low pressure center oriented toward the pole. Regression slopes from the NH are shown in (a-b) and SH slopes are shown in (c-d). (a) and (c) show the slope relating the WCB moisture flux into the cyclone and $LWP_{ij}$ (units are kg mm day$^{-1}$m$^{-2}$). (b) and (d) show the slope of the regression relating $SST_{ij}$ and $LWP_{ij}$ (kg m$^{-2}$K$^{-1}$). The 95% confidence intervals in the zonal-mean regression slope are shown as shading.

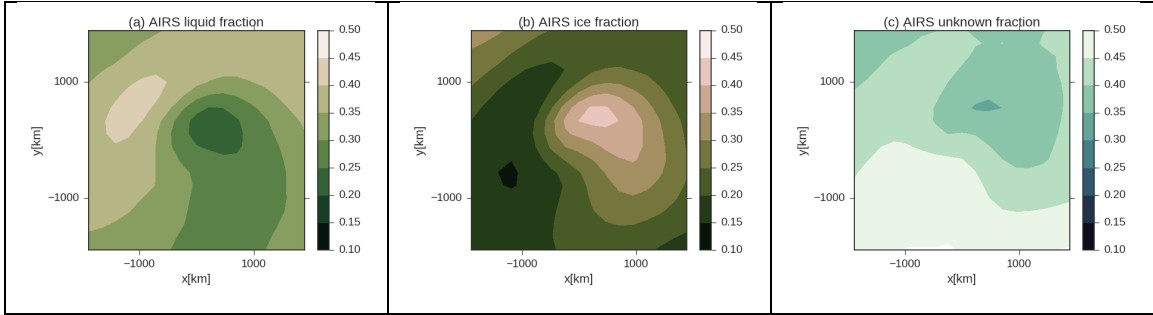

**Fig. 11** Fraction of liquid (a), ice (b), and unknown (c) cloud top phase from AIRS. Fractions are averages over the period 2003-2015 and are for both hemispheres.





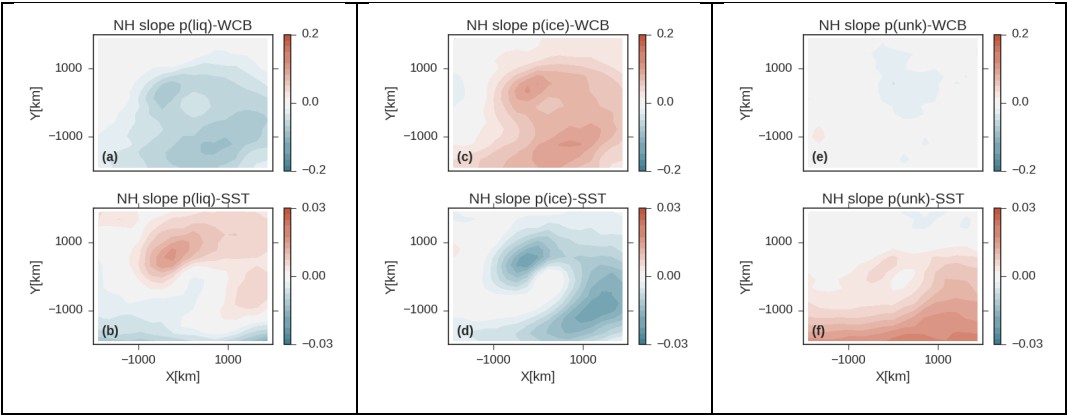

**Fig. 12 as in Fig. 9, but showing the coefficients in the multiple linear regression relating the probability of liquid, ice, and unknown cloud top phase to WCB moisture flux (units are day mm$^{-1}$) into the cyclone and SST$_{ij}$ within the cyclone (units are K$^{-1}$, Eq. 6). All data is from the NH. (a and b) relate to the probability of liquid topped clouds, (c and d) relate to ice-topped clouds, and (e and f) to unknown phase. Note that all probabilities are the probability of detecting a specific phase, given that a phase detection has been made. The first row shows the coefficient relating WCB moisture flux into the cyclone to cloud top phase probability. The second row shows the coefficient between SST$_{ij}$ and cloud top phase.**

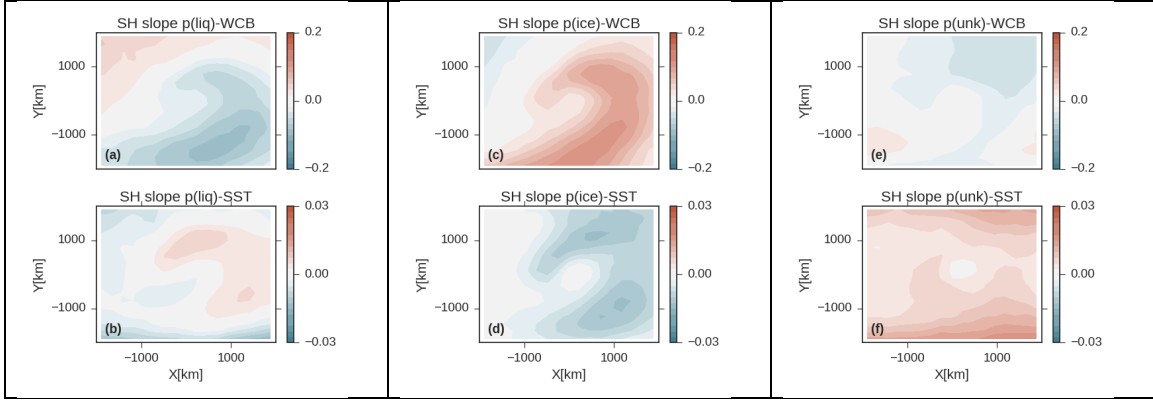

**Fig. 13 as in Fig. 12, but for the SH.**

