# Peer review of "Cloud feedbacks in extratropical cyclones: insight from long-term satellite data and high-resolution global simulations"

_Atmospheric Chemistry and Physics, 2018_

## Referee Comment (RC1) · Anonymous Referee #1 · 18 Sep 2018

The paper aims to put forward an alternative view for mechanisms involved in cloud feedbacks over the extratropical oceans. It promotes the role of increasing water vapour fluxes into cyclones as a mechanism to increase the liquid water path and as a result the reflected shortwave radiation. It attempts to contrast this to the mechanism the commonly held view that it is a phase change from ice to water clouds in a warmer climate that provides a negative cloud feedback over the extratropical oceans.

While the idea is well worth pursuing the paper fails to convince me. There are two reasons. First, I am not convinced the methodology is sufficient to make the case the authors wish to make. Second, the paper is written in a very cumbersome way making

it way too long and a fairly arduous read. The style dilutes many of the arguments the authors wish to put forward with long rambling paragraphs of speculations all throughout the results section.

I will give more details on both points below. I believe the paper can be brought to publication in two ways.

1) Instead of promoting their view of water vapour flux as the alternative to phase changes as the mechanism for feedback, a more measured judgement that both might be at work would be helpful. I believe the methodology used highlights the role of the WCB but cannot exclude the possibility of the phase change hypothesis due to the strong serial correlation of both through temperature. The authors admit this themselves (page 18, line 33). The largest of all correlations in the entire study is that between WCB and SST, making it very hard to argue one way or the other, so why even try. A less strong but equally important conclusion the paper can draw is that it is likely that the WCB effect needs to be considered as a possible, but perhaps not the only, mechanism.

2) The paper needs to be rewritten and significantly shortened. 26 pages of dense text and 28 figures (including the supplement) is simply too much. Many of the supplementary figures are used for major arguments in the text, so they are anything but supplemental.

Major comments

3) The methodology of linear regressions, which is used to make major arguments about processes, is insufficient. Take as an example equation 4. Not only are the two predictors highly correlated, but the physical arguments surrounding them are flawed. Whilst in PBL clouds it is sensible to assume that the cloud temperature is strongly coupled to the SST, I fail to see why this would be true in extratropical cyclones. Furthermore the very old idea the LWP is simply a function of temperature has been discarded for a while now. Not surprisingly then, the method reveals that almost all of

the relationship resides in the first term, which turns out to be mainly due to water vapour increases directly tied to temperature increases, which themselves prohibit you to exclude phase change effects. This highly circular argument makes it very hard to support that rather strong conclusion that it "appears that once WCB moisture flux is accounted for relatively little room is left for an effect related to phase changes." (page 19, line 6). We simply don't know. What we have learned is that phase changes alone might be too simple an explanation. Nothing wrong with that as a conclusion.

4) The averaging over the cyclones is inadequately explained. Page 11, line 10 states that cyclone means are within 2000 km of the cyclone centre. Is this applied to every cyclone? Doesn't it matter how big the cyclone is? Does within mean that sometimes it's less? If the cyclone mean is always 2000 km from the Centre, couldn't this introduce artefacts? If the cyclones are smaller than 2000 km and their size changes, this will change all the averages with little relation to the flux, would it not?

5) The paper needs shortening. This can be achieved in several ways, first and foremost by removing the many long paragraphs of indulgent musings and speculations scattered throughout the results section. They really get in the way of your argument and they should be removed and a short (!) discussion section added after the results instead. Almost every time an interesting result emerges, the reader gets distracted with a paragraph of discussion, sometimes not even strongly related to the results. Sometimes those paragraphs precede new results, making them even more confusing. Here are the most prominent examples for this:

Page 5, Line 5-15: A long paragraph with the figure relegated to the supplementary material. Do we really need this?

Page 14, Line 17-20: What has this to do with what follows? It's just confusing the reader.

Page 15, Line 8-18: Pure speculation. Not a result, so why is it in the results section?

Page 15, Line 23-35: Ditto

Page 16-17, Line 18-5: Again, this has nothing to do with results and simply distracts from them

Page 18, Line 7-8: A very strange sentence. What does this refer to? The un-initiated reader has no idea why this needs yo be discussed. Please revise.

Page 18, Line 11-16: What is this paragraph trying to say? What is it referring to? The previous paper? A figure (7) in this paper? I found it hard to make sense of. Please explain what you are doing, then what your result is. Leave the discussion for a discussion section.

Page 19, Line 12-22: Another distracting paragraph of discussion.

Page 24, Line 1-14: Ditto

6) Another way to shorten the paper is to move the rather detailed model descriptions in Section 2.3 into an appendix.

7) The paper clearly struggles with the use of figures in the supplement. The choice seems almost random and major conclusions are drawn from figures in the supplementary material (as evidenced by many of them mentioned even in the conclusions section). The authors need to revisit all their figures, select the ones that are absolutely necessary for their arguments and omit all others. Which ones will be required will only become clear after the rewrite of the results section, so it is hard to make more concrete suggestions at this stage.

8) The paper switches from global considerations to the SH only and back to global from section to section. I am not sure we learned anything from looking at both hemispheres, so the authors may wish to consider to look at the SH only throughout. This might also tighten the arguments in the paper. One could always include a result from the NH in the discussion section if needed, but keep the main results to one hemisphere.

Minor comments:

Page 5, line 14: You did not state how you composite. Presumably by overlaying the cyclone centers?

Page 5 line 24: The "observations" are presumably a reanalysis - this needs mentioning here. Section 2.2.2: I suggest to move this sentence into the composite section. It is needed there and hardly warrants its on subsection anyway. Also, are daily means good enough to do the cyclone detection? Also, the satellite daily means aren't really daily means. Does this matter? Please discuss this.

Page 7, Line 22-24: Propaganda and not needed here.

Page10, Line 11-13: This is confusing. First you say there is a problem with k, then you use it anyway. Is there a justification for this?

Page 11, Line 28-29: This is a strange sentence. What has societal importance to do with WCB being a useful constraint? Nothing I think. Please change this.

Page 12, Line 21-22: But isn't it the in-cloud LWP that matters? As the dependence is not linear, we could imagine more rain from lower mean but higher in-cloud LWP through changes in cloud fraction.

Page 12, Line 29: The translation to albedo is also non-linear, and more water does not necessarily mean higher albedo. If we are at high LWP where albedo saturates, further increases in LWP will not change albedo. Please discuss this.

Page 14, Line 2-4: In the figure, there aren't many models that flatten more than the observations. On the contrary, there are some that don't flatten at all. So this discussion is one-sided and needs revising.

Page 14, Line 15: What is the "climatology" here? Is it the climatology for each month so as to remove the seasonal cycle? If so, say so.

Page 17, Line 31-32: How can this sentence be true? Are they higher, or are they in

good agreement? They cannot be both!

Page 18 , Line 9: How? Why? Why is any LWP trend equal to a feedback?

Page 20, Line 5: I don't understand this sentence. What does it mean? Why is it there?

Page 20, Line 11: This should be the second sentence of the previous paragraph!

Page 20, Line 26-27: This is and example for a key result with its figure in the supplementary material

---

## Referee Comment (RC2) · Anonymous Referee #2 · 19 Sep 2018

General comments This paper examines the role of the warm conveyor belt (WCB) moisture flux in determining extratropical cyclone variability. It is argued that WCB determines cyclone liquid water path (LWP) variability more than does phase changes in the clouds. Further, as WCB depends on WVP, which will increase under warming according to the Clausius-Clapeyron relation, a negative feedback is identified and quantified. An additional feedback by the second driver of WCB, wind speed, is also discussed, but can't be quantified or even given a certain sign.

The authors are addressing the previously identified negative cloud feedback in the extra tropics, related to cloud optical depth (via LWP), and suggesting a mechanism in

complement or in place of phase changes as responsible for this feedback. This is a valuable contribution.

Comparing a range of model resolutions is a useful approach (although more could be squeezed out of this comparison), as is the cyclone compositing framework.

The way the paper is written, it is somewhat difficult to distill out the main points – a multitude of figures and side tracks make the reasoning hard to follow at times. I would advise the authors to tighten up the writing, and consider reducing the number of figures presented, without simply moving them to the supplementary material. Several of the supplementary figures already play more than a supplementary role, the way the analysis is presently presented.

In addition, the following specific comments will need to be addressed by the authors.

Specific comments 1. The study is based on multiple linear regression (introduced as a statement on p 4, line 22). The authors need to explain why, if at all, this is a suitable approach. It is clear that some of the processes investigated have non-linear elements (e.g. Fig 3). It is also clear that in several cases the predictors are not independent (e.g. Eq. 4, Eq. 6). SST determines WVP through Clausius- Clapeyron, and WVP in turn is part of the definition of WCB, and hence SST and WCB, or "thermodynamics" and "meteorology" (p. 18, line 18-19), can't be separated in this way.

The authors occasionally point at these problems, but further explanation and/or justification would be needed (e.g. p. 14 line 21-24, p. 19, line 3-5). For instance, would it be possible to attempt to estimate parameters for a non-linear relation, rather than forcing a linear fit between LWP and WCB? And would it be an option to use only one predictor rather than two, when they are not independent, as is the case for WCB and SST?

2. It also needs to be acknowledged that the degrees of explanation are in general rather low. E.g. p 14, line 31, Fig. 5. P 1 line 33, states that WCB "can explain" trend

in LWP over two decades, which is a pretty strong statement. P 4 line 13 refers to a "clear criterion" between "synoptic state" (WCB) and LWP to test models against. I find this to be a bit optimistic, based on the results presented. On p 13, line 31-33, it also seems as if the large uncertainty in the observationally based estimate would limit the usefulness of the suggested constraint on models.

Figure 6 shows slopes of relations that (according to Fig. S5) have correlations R2 ranging from below 0.3 to above 0.7. Even though the slopes are all significantly greater than zero, the relations are in some cases rather weak, and a chain of weak correlations is simply not enough to support the conclusions drawn. Could a threshold R2 be used to select a subset of slopes to use?

Another example is p 16, where the reasoning seems to be that latitude explains wind speed which explains WCB which explains precipitation. As stated by the authors, the relation between latitude and windspeed is not causal, but can be explained by poleward travelling and intensification of cyclones during their life cycle. The link to a poleward shift in storm track position is not clear. The change in latitude could leave the initial wind speed unaffected, i.e. the intensification of storms seen is not necessarily an effect of their shift in position.

The weak relations also cause problems in the attempts to compare present climate to future (warmer) conditions. On p 20 it remains unexplained why shifts in the LWP-WCB relation occur and why in the NH (Fig. s9) the shift changes sign between low and high WCB, but it is clear that the assumption that the relationship between WCB and LWP is invariant under warming (p 20 line 4) does in fact not hold, other than within a large range of uncertainty.

3. The paper claims to show that precipitation is balanced by WCB, but I would argue that this is not shown, but rather assumed in Section 3.1., and then used to motivate the continued analysis.

Eq. 3 relates WCB to WVP and WS as WCB=k*WVP*WS. Line 10, however, states

that the constant of proportionality k is defined based on regression of precipitation rate on WVP and WS, i.e. precipitation=k*WVP*WS. This suggests that an equivalence between WCB and precipitation rate is assumed. This is a logical problem (assuming a relation you set out to test) and a physical problem (as there may be a fraction of the precipitation that is not related to the WCB , see e.g. Pfahl et al. 2013, https://journals.ametsoc.org/doi/10.1175/JCLI-D-13-00223.1)

Further down, a "match" between moisture flux into and precipitation out of a cyclone is said to be examined (page 11, line 14-16), and Fig. 2 suggests that models' estimate of the relation between precipitation and WVP*WS is in general agreement with observations (with large uncertainty). It needs to be sorted out what is assumed and what is investigated, in observations and models, and the recurring assumption that WCB can be replaced with precipitation needs explanation and/or justification.

With the current presentation, the statement on P12 Line 16 is not correct; section 3.1 doesn't show that precipitation is predicted by WCB, it shows that WS*WVP is well correlated with (or "predicts") WCB, and it is assumed that WCB is perfectly balanced by precipitation.

On p 13, line 5 it is contrarily stated that the relation that has so far been assumed between WCP and precipitation can't be evaluated. Adding LWP to the discussion here (p 12-13) does not necessarily help. A time aspect seems to be missing, as it is assumed that more moisture flux in is balanced by more precipitation, but in between an observable build-up of liquid water is expected. This requires some explanation.

One option would be to exclude the section on precipitation, and focus the paper on the discussion of the role of WCB in determining LWP.

4. Some other methodological choices are also not clear or explained, e.g. the separation between NH and SH in some cases, and in others not. Please make clear when and why this separation is useful or meaningful. The main focus seems to be on the SH and the Southern Ocean, and perhaps it could be motivated to make that focus

even more distinct.

5. P13, line 8-10 The statement has been reversed, according to Fig. S2 the ratio LWP/TLWP decreases with increasing WCB. This is problematic as it contrasts with the following statement that LWP increases with increasing WCB in general. This does not follow from Fig S2. In Fig .3 the relation between LWP and WCB has the expected sign. On p 14, lines 1-2 a more reasonable interpretation of Fig s2 and fig 3 is made: at higher WCB the partitioning of LWP is more biased towards rain, and this contributes to the asymptotic shape of the LWP-curve in fig 3. I would encourage the suggested future study of this aspect.

6. P16, lines 1-5 (Fig. S3) indicates that a shift of 5 degrees improves the correlation between LWP and WCB. P 18 uses a new choice of latitude range, to agree with Manaster et al. (2107) How region-sensitive might the analysis be, or rather what is the motivation for the chosen regions?

7. It is not clear how the regression of albedo on LWP accounts for cloud masking. (Page 22, line 12 and onward, particularly lines 22 -33 of page 22) It seems like the exercise described here is an attempt to quantify the albedo changes due to changes in LWP due to changes in WCB (or SST), i.e. the suggested feedback mechanism, not to correct for overlying ice clouds. The summarizing sentence on p 23, line 6 also indicates that this is what has been done, rather than accounting for cloud masking

Technical comments Fig1: Observations should be MERRA reanalysis

P3, line 10 "increases or decreases"

P3 line 14 "optical depth increase" should rather read "optical depth change" as it was just stated that it is unclear if it is an increase or a decrease

P3, line 22, line 24 "reflected shortwave" should be followed by radiation

In section 2.1 I would suggest to present the Cyclone compositing (2.1.2) before the Regression analysis (2.1.1), as the compositing is referred to in 2.1.1. but not explained

until 2.1.2.

Section 2.1.2 Cyclone compositing , p5, could use some clarification. E.g. line 9 "before and after" what? please clarify if p_0′ is a function of time or of x,y only. Line 9 "Candidate gridpoints" means what? Line 14 "maximum negative anomaly within 2000 km" of what? Line 17 "of the figure", comes without previous reference to a figure

Page 5, line 28 Please clarify what "bias-corrected" refers to. The following paragraphs describes various identified problems with the MAC LWP data, but it is not clear which if any of these are corrected for, or if excluding certain years and judging surface con-tamination as irrelevant is the bias correction

P6, line 26 please spell out FOV. Throughout, there are many abbreviations, some of which may not be necessary to introduce.

P7 line 25 Please explain Easy Aerosol. As the abbreviations mentioned above, jargon makes the paper more difficult to follow.

P7 line 29 How are these "three resolutions" of HadGEM3 referred to?

P8, line 30 "in more detail"

P10, line 8-9 This statement raises more questions than it answers. I would suggest to explain, or remove it.

P10, line 23 The statement "The January SST was reflected north-south" needs expla-nation

Page 12, line 8: the word models seems to be missing, midlatitude-cyclones can hardly be said to under-estimate precipitation

Page 12 line 10-12 Look over this sentence, it does not make sense

P 13, line 9 fix typo "...results in a decreases the..."

P14 line 31 missing "is"

P14, line 13 "poleward of 30-80N" should be "between 30 and 80N"

P 14 line 25 one "of" too many

P21, line 15-17 please look over this sentence, perhaps removing "that" is all that is needed

P22 line 15 "shortwave radiation"

P22 line 18 "these data"

P24, line 8 "the magnitude"

P24 line 24-26, please look over this sentence

The paper has a somewhat abrupt ending, please consider a final sentence to wrap up. This would be made easier if the whole paper could be more condensed.

---

## Author Comment (AC1) · 23 Nov 2018

The response to reviewers and a track changes version of the revised MS are attached as a PDF.

Please also note the supplement to this comment:
https://www.atmos-chem-phys-discuss.net/acp-2018-785/acp-2018-785-AC1-supplement.pdf

---

## Author Response (AR1)

**Response to reviewers:**

**R1:**

We thank the reviewer for their insightful comments regarding our paper. We acknowledge that the paper is perhaps a little on the long side due to enthusiasm to utilize a range of high-resolution models and exciting new observational data. Because we tried to fit all of this in the reviewer has understandably missed some of the arguments we have made. Following his/her advice we have tightened the paper's structure to better present our arguments. Reviewer comments are in italics.

*1) Instead of promoting their view of water vapour flux as the alternative to phase changes as the mechanism for feedback, a more measured judgement that both might be at work would be helpful. I believe the methodology used highlights the role of the WCB but cannot exclude the possibility of the phase change hypothesis due to the strong serial correlation of both through temperature. The authors admit this them-selves (page 18, line 33). The largest of all correlations in the entire study is that between WCB and SST, making it very hard to argue one way or the other, so why even try. A less strong but equally important conclusion the paper can draw is that it is likely that the WCB effect needs to be considered as a possible, but perhaps not the only, mechanism.*

Please note that throughout the manuscript we do not discount the possibility that the phase transition may be at work – we repeatedly refer to the WCB moisture flux as predicting the majority of the response to transient warming and the decadal trend, not all of it (P1 L33, P4 L7). In fact, the last part of the discussion (section 3.4 of the previous version) centers around investigating the influences of changes in phase using cloud-top phase observations from AIRS.

The reviewer makes a good point that in the context of analysis of covariability we did not sufficiently explain why covariability between SST and WVP via Clausius-Clapeyron isn't a significant issue in inferring the feedback. As the reviewer notes, we were careful to point this out– in fact, it is a central issue in the literature that phase changes and the WCB-driven change in LWP are very easy to conflate if a naïve analysis based on SST alone is pursued. However, the correlation between SST and LWP is very weak (r=0.3 on a cyclone, by cyclone basis). We have added figures showing this. In the original manuscript we looked at dropping each predictor and found that the coefficient relating SST to LWP flipped in sign if WCB was dropped as a predictor (paragraph starting on P18 L32 of the original ms). We have added additional text throughout the manuscript showing that SST is a poor predictor of LWP, even if it is a good predictor of WVP. This is important to thoroughly discuss, especially given that analysis relating LWP to SST alone will give a poor prediction of feedbacks. We thank the reviewer for encouraging us to expand on this.

The reviewer incorrectly points out that our analysis is only based on regression. Our analysis inferring the cloud feedback from the current climate is indeed based on regression. However, following previous work inferring feedbacks from the observational record (Qu et al., 2015) we use modelling simulations of transient warming to test whether these inferences hold when the climate is warmed (AMIP and AMIP+4K). These demonstrate that by using the AMIP WCB-
LWP relationship we can predict the majority of changes in cyclone LWP between AMIP and
AMIP+4K. Because of length of the MS we had moved most of the figures relating to this
analysis to the supplementary material since it seemed like the main result of this section was to
affirm that the relationship between LWP and WCB in the present can predict the future. To
better respond to the reviewer's argument we have rewritten portions of the main text to clarify
our arguments and separated our discussion into sections that compartmentalize this analysis so
that the reader can better follow the argument. Figures that are not essential to the story have
been removed from the main text. Essential figures that were in the SM have been moved to the
main text. The discussion has been refocused on the SH for clarity (equivalent figures for the NH
have been added to the SM).
*2) The paper needs to be rewritten and significantly shortened. 26 pages of dense*
*text and 28 figures (including the supplement) is simply too much. Many of the sup-*
*plementary figures are used for major arguments in the text, so they are anything but*
*supplemental*
As noted in the comment above, we have reorganized and shortened the text to better explain
our argument, although we note that many of the figures are just repetitions of the same figure
for different regions so they are not entirely new figures to digest and we were just showing them
to try and be thorough. We have refocused on the Southern Ocean and moved equivalent NH
figures to the SM.
*3) The methodology of linear regressions, which is used to make major arguments*
*about processes, is insufficient. Take as an example equation 4. Not only are the two*
*predictors highly correlated, but the physical arguments surrounding them are flawed.*
*Whilst in PBL clouds it is sensible to assume that the cloud temperature is strongly*
*coupled to the SST, I fail to see why this would be true in extratropical cyclones. Fur-*
*thermore the very old idea the LWP is simply a function of temperature has been dis-*
*carded for a while now. Not surprisingly then, the method reveals that almost all of the*
*relationship resides in the first term, which turns out to be mainly due to water*
*vapour increases directly tied to temperature increases, which themselves prohibit you*
*to exclude phase change effects. This highly circular argument makes it very hard to*
*support that rather strong conclusion that it "appears that once WCB moisture flux is*
*accounted for relatively little room is left for an effect related to phase changes." (page*
*19, line 6). We simply don't know. What we have learned is that phase changes alone*
*might be too simple an explanation. Nothing wrong with that as a conclusion.*
Please see our response to the reviewer's point one. Our analysis of transient warming
simulations supports our analysis of covariability within the current climate. We have added
additional analysis to the text showing that SST is not a good predictor of cyclone LWP. We
have proposed a simple mechanism based on the moisture flux. We have found that this
mechanism can predict the majority of the observed trend in response to the warming of the
Southern Ocean and can predict the majority of the response to increasing SST by 4K in GCMs.
The SST is not a perfect analog for temperature in cloud, but once you remove the contributions from the WCB moisture flux, there is very little room left over for the mixed-phase cloud
feedback.
*4) The averaging over the cyclones is inadequately explained. Page 11, line 10 states*
*that cyclone means are within 2000 km of the cyclone centre. Is this applied to every*
*cyclone? Doesn't it matter how big the cyclone is? Does within mean that sometimes*
*it's less? If the cyclone mean is always 2000 km from the Centre, couldn't this introduce*
*artefacts? If the cyclones are smaller than 2000 km and their size changes, this will*
*change all the averages with little relation to the flux, would it not?*
The average is the mean within 2000km and is not altered as a function of cyclone structure. This
technique is common in the literature and is explained in the cited papers and in the methods of
this and the previous version of the article (Field et al., 2011;Field et al., 2008;Field and Wood,
2007;Bodas-Salcedo et al., 2016;Bodas-Salcedo et al., 2012;Bodas-Salcedo et al., 2014). It is not
clear what the benefit of a more complex methodology would be in the context of examining the
relationship between WCB moisture flux and LWP in cyclones. To acknowledge that this is not
the only way that cyclone compositing can be achieved we have added the sentence: "More
complex analysis techniques exist that allow the definition of the edge of a cyclonic system(Pfahl
and Sprenger, 2016)."
*5) The paper needs shortening. This can be achieved in several ways, first and fore-*
*most by removing the many long paragraphs of indulgent musings and speculations*
*scattered throughout the results section. They really get in the way of your argument*
*and they should be removed and a short (!) discussion section added after the results*
*instead. Almost every time an interesting result emerges, the reader gets distracted*
*with a paragraph of discussion, sometimes not even strongly related to the results.*
*Sometimes those paragraphs precede new results, making them even more confus-*
*ing. Here are the most prominent examples for this*
We have shortened and rewritten the paper to better showcase our argument. Unfortunately,
there seems to be some issues with the page numbering in the reviewer's copy and it is hard to
follow what page they are looking at. However, we have tried our best to streamline the paper.
*Page 5, Line 5-15: A long paragraph with the figure relegated to the supplementary*
*material. Do we really need this?*
Please note that this is just the methodology for compositing in the paper as shown at:
https://www.atmos-chem-phys-discuss.net/acp-2018-785/acp-2018-785.pdf
*Page 14, Line 17-20: What has this to do with what follows? It's just confusing the*
*reader.*
As noted above, there seems to be an issue with the page numbering in the reviewer's copy. In
the official copy provided by Copernicus this is just a discussion of our choice to use linear
regression instead of the exponential fit used in our previous paper(McCoy et al., 2018).

*Page 15, Line 8-18: Pure speculation. Not a result, so why is it in the results section?*
*Page 15, Line 23-35: Ditto*

It was too weak to start this paragraph with 'It is interesting to speculate.' It is more than speculation. In McCoy et al. (2018) we showed that if cloud droplet number concentration (CDNC) was included as a predictor it substantially increased the variability in cyclone LWP that was explained. The focus of this paper is the WCB moisture flux so we have dropped the CDNC as a predictor so we can compare across more models. However, the difference in explained variance between basins is consistent with variance in CDNC. We have changed the text to clarify that this is not wild speculation and have shortened this section.

*Page 16-17, Line 18-5: Again, this has nothing to do with results and simply distracts*
*from them*
We feel that because our results center on the relationship LWP=a*WCB=a*(k*WVP*WS) we need to at least discuss how WS changes in extratropical cyclones. This is a much more complex question than changes in WVP (which is just Clausius-Clapeyron) so we have added this paragraph to discuss the existing scholarship in the field. In the revised manuscript we have cut down the paragraph for brevity.

*Page 18, Line 7-8: A very strange sentence. What does this refer to? The un-initiated*
*reader has no idea why this needs yo be discussed. Please revise.*
This sentence has been rewritten. It was just to make it clear that a trend within this data set (although not composited) has already been shown in the literature.

*Page 18, Line 11-16: What is this paragraph trying to say? What is it referring to?*
*The previous paper? A figure (7) in this paper? I found it hard to make sense of.*
*Please explain what you are doing, then what your result is. Leave the discussion for a*
*discussion section*

This is referring to Figure 7 of this paper. We have rewritten the paragraph to clarify it further. It shows that the trend in LWP within cyclones across the SH agrees with the trend in the zonal-mean LWP shown in (Manaster et al., 2017).

*Page 19, Line 12-22: Another distracting paragraph of discussion.*
We feel that in order to contrast our result to previous work that is not subset to cyclonic regimes we need to have some discussion of what is happening in anti-cyclones. While we appreciate the reviewer trying to help us streamline our paper, we also worry that other readers within the cloud feedback community will want it clarified how our work fits into the existing literature that is not focused on cyclones.

*Page 24, Line 1-14: Ditto*

Based on comments from this reviewer and reviewer 2 we have focused on the SH in the main text of the paper. However, the difference between the NH and SH cloud-top phase detections in

AIRS necessitate contrasting these regions. We have added some text explaining why we are
making this comparison.
*6) Another way to shorten the paper is to move the rather detailed model descriptions*
*in Section 2.3 into an appendix.*
Thank you- that is a good idea. These have been moved to the supplementary material.
*7) The paper clearly struggles with the use of figures in the supplement. The choice*
*seems almost random and major conclusions are drawn from figures in the supple-*
*mentary material (as evidenced by many of them mentioned even in the conclusions*
*section). The authors need to revisit all their figures, select the ones that are abso-*
*lutely necessary for their arguments and omit all others. Which ones will be required*
*will only become clear after the rewrite of the results section, so it is hard to make more*
*concrete suggestions at this stage*
As discussed in the response to point 1, we have reorganized the paper to remove unnecessary
figures and shift more relevant figures to the main text. The main text focusses on the SH now,
but equivalent figures for the NH have been inserted into the SM.
*8) The paper switches from global considerations to the SH only and back to global*
*from section to section. I am not sure we learned anything from looking at both hemi-*
*spheres, so the authors may wish to consider to look at the SH only throughout. This*
*might also tighten the arguments in the paper. One could always include a result from*
*the NH in the discussion section if needed, but keep the main results to one hemi-*
*sphere.*
This is a good suggestion. We have moved the NH results to the supplementary material as they
support our results in the SH.
*Page 5, line 14: You did not state how you composite. Presumably by overlaying the*
*cyclone centers?*
Data is averaged onto an equal area grid centered on the center of the cyclone- following (Field
and Wood, 2007) as cited in the methodology. We have added a sentence to this effect.
*Page 5 line 24: The "observations" are presumably a reanalysis - this needs mentioning*
*here.*
Changed title to observations and reanalysis.
*Section 2.2.2: I suggest to move this sentence into the composite section. It is*
*needed there and hardly warrants its on subsection anyway. Also, are daily means*
*good enough to do the cyclone detection? Also, the satellite daily means aren't really*
*daily means. Does this matter? Please discuss this.*
To follow the structure of the section and allow the reader to quickly see what data sets we are
using we will keep it as is. It is unclear what the reviewer means by 'good enough'. In (McCoy
et al., 2018) we found reasonable-looking cyclone composites and in that study and the present
study we have found strong relationships between moisture flux and rain rate. It is a good point
that because of its limited overpasses AIRS is not diurnally-averaged as MAC-LWP and CERES

are. This was discussed in the original text (P6 L30), but we have added text further clarifying this to the discussion of cloud-top phase.

"Cloud top phase is measured by the AIRS instrument during the period 2003-2015. It is important to caveat the following analysis by noting that, unlike the other observational data sets used in this paper (MAC-LWP, and CERES), data from AIRS is not diurnally-averaged. It is only available for the Aqua satellite's overpass times. The effects of this temporal subsetting of the data are not clear. However, the goal of the analysis we are pursuing is qualitative. Our intention is to see if liquid cloud phase increases at the expense of ice phase with increasing SST in the same regions that LWP increases with increasing SST. Fig. S15 shows cyclone composited AIRS observations. The structure of ice and liquid phase exhibits a reasonable ice cloud shield and liquid warm sector- indicating that it may shed at least some light on variability in cloud-top phase within cyclones."

*Page 7, Line 22-24: Propaganda and not needed here.*
 Removed.

*Page10, Line 11-13: This is confusing. First you say there is a problem with k, then*
*you use it anyway. Is there a justification for this?*
These line numbers refer to the UM-CASIM model description, but since your question is generally about using k this seems to be about page 11. We go on to discuss the actual k from the models and the range of k's consistent with the observations. We have added the sentence, 'for consistency with previous literature we have chosen the k based on AMSR-E observations. Our results might change slightly in a quantitative sense if another k was used, but will remain qualitatively the same.'.

*Page 11, Line 28-29: This is a strange sentence. What has societal importance to do*
*with WCB being a useful constraint? Nothing I think. Please change this.*
The sentence has been reordered to make it clear that societal importance does not make it a key constraint on precipitation. Precipitation is of societal importance and thus WCB moisture flux is a key constraint on the climate.

*Page 12, Line 21-22: But isn't it the in-cloud LWP that matters? As the dependence*
*is not linear, we could imagine more rain from lower mean but higher in-cloud LWP*
*through changes in cloud fraction.*
Neither the rain rate nor the LWP are in-cloud. We have added some text clarifying this.

*Page 12, Line 29: The translation to albedo is also non-linear, and more water does*
*not necessarily mean higher albedo. If we are at high LWP where albedo saturates,*
*further increases in LWP will not change albedo. Please discuss this.*
We discussed this at length in our preceding paper(McCoy et al., 2018) as cited on P12 L24 of the original text, but we neglected to add a citation here. We have added a citation to this. Thanks.

*Page 14, Line 2-4: In the figure, there aren't many models that flatten more than the ob-*
*servations. On the contrary, there are some that don't flatten at all. So this discussion*

*is one-sided and needs revising.*
We have added 'and vice-versa' at the end of the sentence.
*Page 14, Line 15: What is the "climatology" here? Is it the climatology for each month*
*so as to remove the seasonal cycle? If so, say so*
It has been noted that the climatology is a monthly-mean climatology.
*Page 17, Line 31-32: How can this sentence be true? Are they higher, or are they in good*
*agreement? They cannot be both!*
This has been expanded to clarify this.
*Page 18 , Line 9: How? Why? Why is any LWP trend equal to a feedback?*
Because there is a steady warming signal, a trend over time is probably a response to warming.
We have added a sentence to clarify this.
*Page 20, Line 5: I don't understand this sentence. What does it mean? Why is it there?*
Because we only have AMIP+4K simulations for some of the models. We wanted to clarify why
this was.  The PRIMAVERA simulations are very expensive and slow to run and the warming
simulations will not be done for quite some time. We realize that it is not entirely clear to the
reader that there was another part to this paper (testing the predictions from the current climate's
behavior in a simulation of a warmed climate, see major comments) and we have reorganized to
clarify this. We have also restructured the sections to allow the reader to see the different parts of
our argument more clearly.
*Page 20, Line 11: This should be the second sentence of the previous paragraph!*
It seems more appropriate to leave this sentence where it is as it forms a topic sentence for the
following paragraph.
*Page 20, Line 26-27: This is and example for a key result with its figure in the supple-*
*mentary material*
As noted above, we have reorganized the paper to make our analysis in the AMIP+4K
simulations clearer.
Bodas-Salcedo, A., Williams, K. D., Field, P. R., and Lock, A. P.: The Surface Downwelling
Solar Radiation Surplus over the Southern Ocean in the Met Office Model: The Role of
Midlatitude Cyclone Clouds, Journal of Climate, 25, 7467-7486, 10.1175/jcli-d-11-00702.1,
2012.
Bodas-Salcedo, A., Williams, K. D., Ringer, M. A., Beau, I., Cole, J. N. S., Dufresne, J. L.,
Koshiro, T., Stevens, B., Wang, Z., and Yokohata, T.: Origins of the Solar Radiation Biases over
the Southern Ocean in CFMIP2 Models, Journal of Climate, 27, 41-56, 10.1175/jcli-d-13-
00169.1, 2014.

Bodas-Salcedo, A., Andrews, T., Karmalkar, A. V., and Ringer, M. A.: Cloud liquid water path and radiative feedbacks over the Southern Ocean, Geophys. Res. Lett., n/a-n/a, 10.1002/2016GL070770, 2016.

Field, P. R., and Wood, R.: Precipitation and cloud structure in midlatitude cyclones, Journal of Climate, 20, 233-254, 10.1175/jcli3998.1, 2007.

Field, P. R., Gettelman, A., Neale, R. B., Wood, R., Rasch, P. J., and Morrison, H.: Midlatitude Cyclone Compositing to Constrain Climate Model Behavior Using Satellite Observations, Journal of Climate, 21, 5887-5903, doi:10.1175/2008JCLI2235.1, 2008.

Field, P. R., Bodas-Salcedo, A., and Brooks, M. E.: Using model analysis and satellite data to assess cloud and precipitation in midlatitude cyclones, Quarterly Journal of the Royal Meteorological Society, 137, 1501-1515, 10.1002/qj.858, 2011.

Manaster, A., O'Dell, C. W., and Elsaesser, G.: Evaluation of Cloud Liquid Water Path Trends Using a Multidecadal Record of Passive Microwave Observations, Journal of Climate, 30, 5871-5884, 10.1175/jcli-d-16-0399.1, 2017.

McCoy, D. T., Field, P. R., Schmidt, A., Grosvenor, D. P., Bender, F. A. M., Shipway, B. J., Hill, A. A., Wilkinson, J. M., and Elsaesser, G. S.: Aerosol midlatitude cyclone indirect effects in observations and high-resolution simulations, Atmospheric Chemistry and Physics, 18, 5821-5846, 10.5194/acp-18-5821-2018, 2018.

Qu, X., Hall, A., Klein, S. A., DeAngelis, and Anthony, M.: Positive tropical marine low-cloud cover feedback inferred from cloud-controlling factors, Geophys. Res. Lett., n/a-n/a, 10.1002/2015GL065627, 2015.

**R2:**

*The authors are addressing the previously identified negative cloud feedback in the extra tropics, related to cloud optical depth (via LWP), and suggesting a mechanism in complement or in place of phase changes as responsible for this feedback. This is a valuable contribution.*

*Comparing a range of model resolutions is a useful approach (although more could be squeezed out of this comparison), as is the cyclone compositing framework.*

*The way the paper is written, it is somewhat difficult to distill out the main points – a multitude of figures and side tracks make the reasoning hard to follow at times. I would advise the authors to tighten up the writing, and consider reducing the number of figures presented, without simply moving them to the supplementary material. Several of the supplementary figures already play more than a supplementary role, the way the analysis is presently presented.*

We thank the reviewer for their thorough appraisal of our paper and supportive feedback. In the process of writing the paper we got slightly carried away with the exciting new range of high-resolution simulations and observational data sets available. We have worked to streamline the paper and make it punchier and to make the central points of the analysis clearer. We have also restructured the sections so that our line of argument becomes clearer and refocused our analysis on the SH.

*Specific comments 1. The study is based on multiple linear regression (introduced as a statement on p 4, line 22). The authors need to explain why, if at all, this is a suitable*

*approach. It is clear that some of the processes investigated have non-linear elements*
*(e.g. Fig 3). It is also clear that in several cases the predictors are not independent*
*(e.g. Eq. 4, Eq. 6). SST determines WVP through Clausius- Clapeyron, and WVP in*
*turn is part of the definition of WCB, and hence SST and WCB, or "thermodynamics"*
*and "meteorology" (p. 18, line 18-19), can't be separated in this way.*

As discussed in the response to R1, we used the AMIP+4K simulations to justify the use of
predictions based on linear-regression in the current climate. This is following previous literature
supporting inference of cloud feedbacks from the observational record with model
simulations(Qu et al., 2015). However, this was not sufficiently clearly presented in the original
paper and we have tried to clarify this. The reviewer makes a good point that the statement on
page 4 is misleading. We have altered it to clarify that regression analysis is only part of our
analysis.

The reviewer's point that WCB moisture flux is going to contain a significant thermodynamic
component (eg through Clausius-Clapeyron) is very true and this discussion has been changed to
reflect this throughout the paper. We have also added additional discussion of SST as a predictor
of LWP and analysis showing that SST and LWP are poorly correlated.

On p.19 line 3-5 we also examined the linear regression on only one predictor at a time
(importantly the dependence on SST flips, which is not consistent with simply sharing
covariability and both predictors being equally good). We have added new analysis showing the
generally poor correspondence between SST and LWP (Fig. 6 of the new MS and Fig. S7).

We believe that we have pushed the regression analysis in the current climate as far as we really
can and that there is not evidence that covariability between WVP and SST degrades WCB
moisture flux as a predictor. Further, we find that this regression model does a good job at
predicting the majority of the transient climate response in models.

*The authors occasionally point at these problems, but further explanation and/or justi-*
*fication would be needed (e.g. p. 14 line 21-24, p. 19, line 3-5). For instance, would*
*it be possible to attempt to estimate parameters for a non-linear relation, rather than*
*forcing a linear fit between LWP and WCB? And would it be an option to use only one*
*predictor rather than two, when they are not independent, as is the case for WCB and*
*SST?*
In figures S7 and S8 of the previous version of the paper we utilized a non-linear regression
model on WCB alone to look at predicted changes in LWP. Overall this does not produce a
substantially different prediction of the change in LWP to the simple linear model. We have
added an additional calculation contrasting a linear fit to WCB and find that it does not alter the
prediction of WCB-driven changes in LWP (Fig. 6 of the new MS). We have also added a
calculation based on SST alone (also Fig. 6).

*2. It also needs to be acknowledged that the degrees of explanation are in general*
*rather low. E.g. p 14, line 31, Fig. 5. P 1 line 33, states that WCB "can explain" trend in LWP*
*over two decades, which is a pretty strong statement. P 4 line 13 refers to a*

*"clear criterion" between "synoptic state" (WCB) and LWP to test models against. I find*
*this to be a bit optimistic, based on the results presented. On p 13, line 31-33, it also*
*seems as if the large uncertainty in the observationally based estimate would limit the*
*usefulness of the suggested constraint on models*
*Figure 6 shows slopes of relations that (according to Fig. S5) have correlations R2*
*ranging from below 0.3 to above 0.7. Even though the slopes are all significantly*
*greater than zero, the relations are in some cases rather weak, and a chain of weak*
*correlations is simply not enough to support the conclusions drawn. Could a threshold*
*R2 be used to select a subset of slopes to use?*

'Can explain' might have been somewhat vague. The 95% confidence on the climate trend in the zonal mean from Manaster et al. (2017) is shown in Fig. 7b. We show the 95% confidence trend in cyclone LWP, and the 95% confidence trend predicted by WCB alone. These trends are all significant at this confidence interval and the interval is small. We have added some additional clarification on this statement to clarify what we mean in this case.

In regards to testing models against the observations, we show the 95% confidence on the WCB-LWP relationship in fig 6a. The range of WCB-LWP relationships that are within the observational range is quite small, and several of the models considered here fall significantly outside of it. In an objective observational sense this is our definition of a clear criterion. It is true that the $R^2$ values within the observations are 30-40%. Focusing on this is somewhat misleading. What we are most interested in in this case is the confidence interval on the trend. We have also added discussion of why the explained variance is low- for example, we neglected cloud droplet number concentration as a predictor, which had been shown in McCoy et al. (2018) to significantly increase explained variance. However, if CDNC stays approximately constant during the observational period, or over a transient warming (as it does in the AMIP-AMIP+4K), then this explained variance is unimportant to our ability to understand the climate response. The reviewer makes a very good point that our discussion is not sufficiently clear in regards to what our expectations are for the regression model and we have added some text to clarify this. We have also updated Fig. 2 of the new version of the paper to better show the confidence on the slope of the regression.

*Another example is p 16, where the reasoning seems to be that latitude explains wind*
*speed which explains WCB which explains precipitation. As stated by the authors,*
*the relation between latitude and windspeed is not causal, but can be explained by*
*poleward travelling and intensification of cyclones during their life cycle. The link to a*
*poleward shift in storm track position is not clear. The change in latitude could leave the*
*initial wind speed unaffected, i.e. the intensification of storms seen is not necessarily*
*an effect of their shift in position.*

As shown in Fig. S12 of the original MS the poleward shift within models predicts 86% of the change in wind speed. We do not focus on this relationship within the paper because we are not confident in explaining the causal link, but it does appear to be a robust feature of the warmed climate response and the climatological variability. We have moved Fig. S12 to the main text to clarify this argument. We have added additional text explaining that the shift in mean storm position is likely to indicate some basic change in the cyclone lifecycle in response to warming
and that we are reserving trying to better understand this linkage for a future paper.
*The weak relations also cause problems in the attempts to compare present climate to*
*future (warmer) conditions. On p 20 it remains unexplained why shifts in the LWP-WCB*
*relation occur and why in the NH (Fig. s9) the shift changes sign between low and high*
*WCB, but it is clear that the assumption that the relationship between WCB and LWP*
*is invariant under warming (p 20 line 4) does in fact not hold, other than within a large*
*range of uncertainty.*
As shown in Fig S10 and S11 the majority of the LWP change between AMIP and AMIP+4K
simulations can be calculated based on the current climate's WCB-LWP relationship and the
WCB moisture flux change between AMIP and AMIP+4K. The assumption is that this shift may
be related to any other changes in the clouds with warming (for example a phase transition). We
have added some text to clarify that we are not assuming that the relationship holds. We are
testing how much the relationship can predict. To clarify this we have added these sentences:
"This analysis tests the assumption that the relationship between WCB moisture flux and
LWP is invariant under warming."
*3. The paper claims to show that precipitation is balanced by WCB, but I would argue*
*that this is not shown, but rather assumed in Section 3.1., and then used to motivate*
*the continued analysis.*
*Eq. 3 relates WCB to WVP and WS as WCB=k*WVP*WS. Line 10, however, states that the*
*constant of proportionality k is defined based on regression of precipitation*
*rate on WVP and WS, i.e. precipitation=k*WVP*WS. This suggests that an equiv-*
*alence between WCB and precipitation rate is assumed. This is a logical problem*
*(assuming a relation you set out to test) and a physical problem (as there may be a*
*fraction of the precipitation that is not related to the WCB , see e.g. Pfahl et al. 2013,*
*https://journals.ametsoc.org/doi/10.1175/JCLI-D-13-00223.1)*
*Further down, a "match" between moisture flux into and precipitation out of a cyclone is*
*said to be examined (page 11, line 14-16), and Fig. 2 suggests that models' estimate*
*of the relation between precipitation and WVP*WS is in general agreement with obser-*
*vations (with large uncertainty). It needs to be sorted out what is assumed and what is*
*investigated, in observations and models, and the recurring assumption that WCB can*
*be replaced with precipitation needs explanation and/or justification.*
*With the current presentation, the statement on P12 Line 16 is not correct; section 3.1*
*doesn't show that precipitation is predicted by WCB, it shows that WS*WVP is well*
*correlated with (or "predicts") WCB, and it is assumed that WCB is perfectly balanced*
*by precipitation.*
*On p 13, line 5 it is contrarily stated that the relation that has so far been assumed*
*between WCP and precipitation can't be evaluated. Adding LWP to the discussion*
*here (p 12-13) does not necessarily help. A time aspect seems to be missing, as it is*
*assumed that more moisture flux in is balanced by more precipitation, but in between*
*an observable build-up of liquid water is expected. This requires some explanation.*
*One option would be to exclude the section on precipitation, and focus the paper on*

*the discussion of the role of WCB in determining LWP.*
The reviewer is correct that this was presented as being shown and it should have been presented
as being a predictor-predictand relationship. Overall, our results are insensitive to this. The
paper has been updated to reflect that we find that WCB moisture flux is generally in good
agreement with precipitation out of the cyclone, but we cannot show that it is balanced by it.
*4. Some other methodological choices are also not clear or explained, e.g. the sepa-*
*ration between NH and SH in some cases, and in others not. Please make clear when*
*and why this separation is useful or meaningful. The main focus seems to be on the*
*SH and the Southern Ocean, and perhaps it could be motivated to make that focus even more*
*distinct.*
The reviewer makes a good point that the analysis was somewhat unfocused. We have refocused
the manuscript to examine the SH. The SH is of particular interest because of the large model-
predicted negative cloud feedback in this region and the observed trends(Manaster et al., 2017).
However, we want to note that the relationships we find in the SH are for the most part replicated
in the NH and this material has been moved to the SM in case readers want to satisfy themselves
that this is not a phenomena that is specific to the SH. The exception to only showing the SH in
the new MS is the analysis of AIRS cloud top phase. AIRS detects a very large amount of
unknown-topped cloud in the SH. We discuss why this is likely to happen and possible changes
to the retrieval algorithm that might improve this. However, to be able to say something useful
about how LWP changes and phase changes might be related in extratropical cyclones we found
it useful to include NH and SH observations. We have added this text to explain why we are
doing this:
"In this work we have focused on the SH for brevity because it is interesting from a
modelling perspective and because the behavior of cyclone LWP as a function of WCB moisture
flux in the NH is approximately the same, giving little additional explanatory value to including
it. However, the preponderance of unknown-topped cloud observed by AIRS in the SH
necessitates contrasting NH and SH midlatitude oceans to offer insight into whether cloud top
phase changes might explain some of the response of LWP to SST within cyclones."
*5. P13, line 8-10 The statement has been reversed, according to Fig. S2 the ratio*
*LWP/TLWP decreases with increasing WCB. This is problematic as it contrasts with*
*the following statement that LWP increases with increasing WCB in general. This does*
*not follow from Fig S2. In Fig .3 the relation between LWP and WCB has the expected*
*sign. On p 14, lines 1-2 a more reasonable interpretation of Fig s2 and fig 3 is made: at*
*higher WCB the partitioning of LWP is more biased towards rain, and this contributes*
*to the asymptotic shape of the LWP-curve in fig 3. I would encourage the suggested*
*future study of this aspect.*
Thank you, this was typo in this sentence. We have shifted the material on p14 line 1-2 to
explain this here instead of ending with this statement in the section.
We also think this is a very interesting feature and are working toward evaluating it in a
perturbed parameter ensemble in the UM. Thank you for clarifying this statement.
*6. P16, lines 1-5 (Fig. S3) indicates that a shift of 5 degrees improves the correlation*

*between LWP and WCB. P 18 uses a new choice of latitude range, to agree with*
*Manaster et al. (2107) How region-sensitive might the analysis be, or rather what is*
*the motivation for the chosen regions?*
The region choice of 30-80° was based on the compositing technique used in Field and Wood
(2007). However, we suspect that cyclones near 30° are undergoing the tropical-extratropical
transition and the moisture flux-LWP mechanism that we propose is less relevant to these
cyclones. To support this statement we showed the maps with a 5° shift. We felt it would be
disingenuous to select a latitude region that gave a higher explained variance so we stuck with
30-80°. Overall, the WCB-LWP relationship's slope does not change so our proposed climate
feedback is not sensitive to this choice- even if the inclusion of transitioning cyclones adds some
noise to the relationship. In order to compare to Manaster et al. (2017) we had to select a similar
latitude range to look at the decadal trend. We have removed this aside because it doesn't add
much and distracts from the flow of the paper.
The following sentence has been added to the methods to explain our choice of 30-80°
"For consistency with previous studies utilizing the Field and Wood (2007) cyclone compositing
algorithm cyclone centers must have their center between 30° and 80° latitude."
*7. It is not clear how the regression of albedo on LWP accounts for cloud masking.*
*(Page 22, line 12 and onward, particularly lines 22 -33 of page 22) It seems like the*
*exercise described here is an attempt to quantify the albedo changes due to changes*
*in LWP due to changes in WCB (or SST), i.e. the suggested feedback mechanism,*
*not to correct for overlying ice clouds. The summarizing sentence on p 23, line 6 also*
*indicates that this is what has been done, rather than accounting for cloud masking*
The logic behind this back of the envelope calculation was that the CERES albedo will include
contributions to the optical depth from overlying ice cloud.  If, for example, all cyclones had an
infinitely thick ice-cloud over the liquid cloud then the sensitivity of top of atmosphere albedo to
microwave LWP would be zero (the LWP could do whatever it liked beneath the ice cloud). As
the reviewer says, the goal of this exercise is to quantify the change in albedo consistent with a
change in LWP driven by a change in WCB or SST. However, to do that we need to account for
overlying ice cloud somehow, otherwise our feedback would be unrealistically large.  We have
added some additional text to try and clarify this:
"In particular, overlying cloud can act to blunt the effect of changes in LWP on top of
atmosphere reflected shortwave radiation. For example, an optically thick layer of ice cloud over
the liquid in the cyclone would result in very little impact from LWP variability.  We offer an
approximate calculation of the change in reflected shortwave radition consistent with the
coefficients calculated in Eq. 6 using observations from CERES. The idea underlying this
calculation is that the CERES top of atmosphere reflected shortwave radiation will include the
effects of overlying ice cloud. The sensitivity in reflected shortwave radiation to LWP will be
lowered by the effects of ice cloud."
*Technical comments Fig1: Observations should be MERRA reanalysis*
Changed
*P3, line 10 "increases or decreases"*

Changed

*P3 line 14 "optical depth increase" should rather read "optical depth change" as it was*
*just stated that it is unclear if it is an increase or a decrease*
Changed

*P3, line 22, line 24 "reflected shortwave" should be followed by radiation*
Changed

*In section 2.1 I would suggest to present the Cyclone compositing (2.1.2) before the*
*Regression analysis (2.1.1), as the compositing is referred to in 2.1.1. but not explained until*
*2.1.2.*
Thank you- that is clearer.

*Section 2.1.2 Cyclone compositing , p5, could use some clarification. E.g. line 9*
*"before and after" what? please clarify if p_0' is a function of time or of x,y only. Line 9*
*"Candidate gridpoints" means what? Line 14 "maximum negative anomaly within 2000*
*km" of what? Line 17 "of the figure", comes without previous reference to a figure*
Clarification has been added. We tried to shorten this section as it reproduces the original text in
Field and Wood (2007).

*Page 5, line 28 Please clarify what "bias-corrected" refers to. The following paragraphs*
*describes various identified problems with the MAC LWP data, but it is not clear which*
*if any of these are corrected for, or if excluding certain years and judging surface con-*
*tamination as irrelevant is the bias correction*
Thank you- this needed to be clarified. We have added the following:
"Bias correction was performed using observations from Aqua MODIS. As a function of WVP
and 10-m surface wind, Aqua MODIS was used to determine clear-sky (here, by definition, LWP
= 0) scenes, and these scenes were compared to AMSR-E LWP.  If a non-zero difference was
computed between AMSR-E and MODIS LWP, this difference was removed from all individual
input LWP records (as a function of WVP/wind) prior to processing in the MAC algorithm.  This
LWP bias correction is discussed in more detail in Elsaesser et al. (2017)."

*P6, line 26 please spell out FOV. Throughout, there are many abbreviations, some of*
*which may not be necessary to introduce.*
Changed

*P7 line 25 Please explain Easy Aerosol. As the abbreviations mentioned above, jargon*
*makes the paper more difficult to follow.*
Thank you, a citation to the relevant paper has been added.

*P7 line 29 How are these "three resolutions" of HadGEM3 referred to?*

We have added a note that they are referred to as LM, MM, and HM.

*P8, line 30 "in more detail"*
Changed

*P10, line 8-9 This statement raises more questions than it answers. I would suggest to*

*explain, or remove it*
Removed- it is irrelevant in this case.
*P10, line 23 The statement "The January SST was reflected north-south" needs expla-*
*nation*
This has been expanded for clarity.
*Page 12, line 8: the word models seems to be missing, midlatitude-cyclones can hardly*
*be said to under-estimate precipitation*
Very true- thank you.
*Page 12 line 10-12 Look over this sentence, it does not make sense*
Thank you- there was a typo.
*P 13, line 9 fix typo "...results in a decreases the..."*
Thank you- fixed
*P14 line 31 missing "is"*
Fixed
*P14, line 13 "poleward of 30-80N" should be "between 30 and 80N"*
Fixed
*P 14 line 25 one "of" too many*
Fixed
*P21, line 15-17 please look over this sentence, perhaps removing "that" is all that is*
*Needed*
Fixed
*P22 line 15 "shortwave radiation"*
Fixed
*P22 line 18 "these data"*
Fixed
*P24, line 8 "the magnitude"*
Fixed
*P24 line 24-26, please look over this sentence*
This has been removed to shorten the paper.
*The paper has a somewhat abrupt ending, please consider a final sentence to wrap*
*up. This would be made easier if the whole paper could be more condensed.*
We have tried to rewrite the paper for clarity. Thank you for your advice.
We have changed the conclusion section to have a more rounded discussion and a better ending.
Thank you again for your insight.

[revised manuscript text omitted]

---

## Author Response (AR2)

We thank the editor and reviewers for their contributions to improving the paper. We have made the minor grammatical changes as requested by reviewer 1. These are attached in track changes below.

[revised manuscript text omitted]